# Desmosomal connectomics of all somatic muscles in an annelid larva

**Sanja Jasek[1], Csaba Verasztó[1†], Emelie Brodrick[1], Réza Shahidi[1], Tom Kazimiers[2,3], Alexandra Kerbl[1], Gáspár Jékely[1]\***

[1]Living Systems Institute, University of Exeter, Exeter, United Kingdom; [2]Janelia Research Campus, Ashburn, United States; [3]kazmos GmbH, Dresden, Germany

**Abstract** Cells form networks in animal tissues through synaptic, chemical, and adhesive links. Invertebrate muscle cells often connect to other cells through desmosomes, adhesive junctions anchored by intermediate filaments. To study desmosomal networks, we skeletonised 853 muscle cells and their desmosomal partners in volume electron microscopy data covering an entire larva of the annelid *Platynereis*. Muscle cells adhere to each other, to epithelial, glial, ciliated, and bristle-producing cells and to the basal lamina, forming a desmosomal connectome of over 2000 cells. The aciculae – chitin rods that form an endoskeleton in the segmental appendages – are highly connected hubs in this network. This agrees with the many degrees of freedom of their movement, as revealed by video microscopy. Mapping motoneuron synapses to the desmosomal connectome allowed us to infer the extent of tissue influenced by motoneurons. Our work shows how cellular-level maps of synaptic and adherent force networks can elucidate body mechanics.

## Editor's evaluation

This paper is based on the digital reconstruction of a serial EM stack of a larva of the annelid *Platynereis* and presents a complete 3D map of all desmosomes between somatic muscle cells and their attachment partners. This resource is of interest to scientists in several fields: motor control, high-resolution anatomy, and network analyses. With the first comprehensive and complete mapping of muscle-to-body connectivity through desmosomes in an annelid larva, it has the potential to close a missing link and make progress towards understanding in a "holistic" way how a complex neural circuitry controls an equally complex pattern of movement/behavior.

**\*For correspondence:**
g.jekely@exeter.ac.uk

**Present address:** [†]EPFL Campus Biotech, Geneva, Switzerland

## Introduction

Animals use somatic muscles to rapidly change body shape. Whole-body shape changes can drive swimming, crawling, or other forms of movement. Moving appendages with exo- or endoskeletons and joints can have many degrees of freedom to support walking, flying, and several other functions. This requires a large diversity of muscles with differential nervous control (*Brierley et al., 2012*; *Charles et al., 2016*; *Cruce, 1974*; *Landmesser, 1978*; *McKellar et al., 2020*).

To exert force and induce movements, contracting muscles must attach to other body parts. In many animals, muscle–force coupling is provided by desmosomes – stable adhesive junctions anchored by intracellular intermediate filaments. Desmosomes occur in cardiac and somatic muscles of both vertebrates and invertebrates. In the zebrafish heart for example, desmosomes connect cardiac myocytes to each other (*Lafontant et al., 2013*). In the lamprey, desmosomes frequently unite adjacent muscle fibres for lateral force transmission (*Nakao, 1975*). In the nematode *Caenorhabditis elegans*, hemidesmosomes link the pharyngeal muscles to the basement membrane (*Albertson and Thomson, 1976*) and the body-wall muscles to hypodermal cells that lie underneath the cuticle (*Francis and Waterston,*

*1991*). In leech, muscle cells have a high density of hemidesmosomes anchoring the cells to the basal lamina (*Pumplin and Muller, 1983*). In polychaete annelids, hemidesmosomes and desmosomes are common on the extracellular matrix connecting muscles and epithelial cells (*Purschke, 1985*). In Echiura, hemidesmosomes connect the chaetal follicle cells to the extracellular matrix. Opposite these sites, muscle cells have dense plaques also linking them to the extracellular matrix (*Tilic et al., 2015*).

These desmosomal connections form a body-wide network through which tensile forces propagate. This network comprises all muscles and all their cellular and non-cellular (e.g. basal lamina) partners. We refer to this network as the desmosomal connectome, with muscles, other cells and basal lamina chunks as nodes and hemidesmosomes and desmosomes as links. How such networks are organised on the scale of the whole body in animals with complex muscles and appendages is not known. While whole-body neuronal connectomes (where links are synaptic connections) exist (*Cook et al., 2019*; *Ryan et al., 2016*; *Verasztó et al., 2020*; *White et al., 1986*), to our knowledge, no whole-body desmosomal connectome (where links are desmosomes) has yet been described. Unlike synapses, desmosomal connections are not directional and do not represent signal flow. Rather they indicate physical coupling where the movement of one cell will result in the movement of the connected structure.

To study the organisation of a body-wide tensile network and its motor control, here we reconstruct the desmosomal connectome of larval *Platynereis dumerilii* from serial electron microscopy images (volume EM) of an entire larva (*Randel et al., 2015*). *P. dumerilii* or 'the nereid' is a marine annelid worm that is increasingly used as a laboratory animal (*Özpolat et al., 2021*).

In *Platynereis*, muscle development starts in the planktonic, ciliated trochophore larval stage (1–2 days old) (*Fischer et al., 2010*). The older (around 3–6 days) nectochaete larvae have three main trunk segments, each with a pair of appendages called parapodia. The parapodia are composed of a ventral lobe (neuropodium) and a dorsal lobe (notopodium) and each lobe contains a single acicula and a bundle of chitin bristles (chaetae) (*Hausen, 2005*). *Platynereis* larvae have an additional cryptic segment between the head and the main trunk segments (*Steinmetz et al., 2011*). This cryptic segment – also referred to as segment 0 – lacks parapodia and gives rise to the anterior pair of tentacular cirri that start to grow around 3 days of development.

Nectochaete larvae can either swim with cilia (*Jékely et al., 2008*) or crawl on the substrate by muscles (*Conzelmann et al., 2013*; *Lauri et al., 2014*). Trunk muscles also support turning in swimming larvae during visual phototaxis (*Randel et al., 2014*) and mediate a startle response characterised by whole-body contraction and the elevation of the parapodia (*Bezares-Calderón et al., 2018*). Older (>6-day-old) feeding juvenile stages are mainly crawling and develop a muscular, eversible proboscis (*Fischer et al., 2010*), a moving gut with visceral muscles (*Brunet et al., 2016*; *Williams et al., 2015*), and several sensory appendages (palps, antennae, and cirri).

*Platynereis* larvae (1- to 6-day-old stage) are 150–300 μm long and thus amenable to whole-body volume-EM analysis. Here, we use an available volume-EM dataset of a 3-day-old nectochaete larva (*Randel et al., 2015*) to reconstruct the desmosomal connectome and analyse how motoneurons innervate muscles.

## Results

### Serial EM reconstruction of all muscle cells in the *Platynereis* larva

To analyse the organisation of the somatic musculature, we used a previously reported whole-body serial TEM dataset of a 3-day-old *P. dumerilii* larva (*Randel et al., 2015*). First, we reconstructed all differentiated muscle cells by tracing along their longitudinal axis (skeletonisation) and marking the position of their soma (*Figure 1*, *Figure 2*). We also skeletonised all other cells in the larva, including neuronal and non-neuronal cells (see below and *Verasztó et al., 2020*).

We identified 853 mononucleated muscle cells in the whole-body volume. Most of them have an obliquely striated ultrastructure, with the exception of developing muscle cells (which contain few myosin filaments), and the two MUSmed_head cells which have disorganised myosin filaments. The majority of muscle cells in the 3-day-old larva are somatic muscles (*Table 1*). Most visceral muscles, including gut muscles, develop at a later stage (*Brunet et al., 2016*). Based on their position and shape, we classified all muscle cells into eight main groups (*Figure 2*, *Table 1*): longitudinal, anterior and posterior oblique, chaetal, acicular, transverse, digestive system (developing), and head muscles.

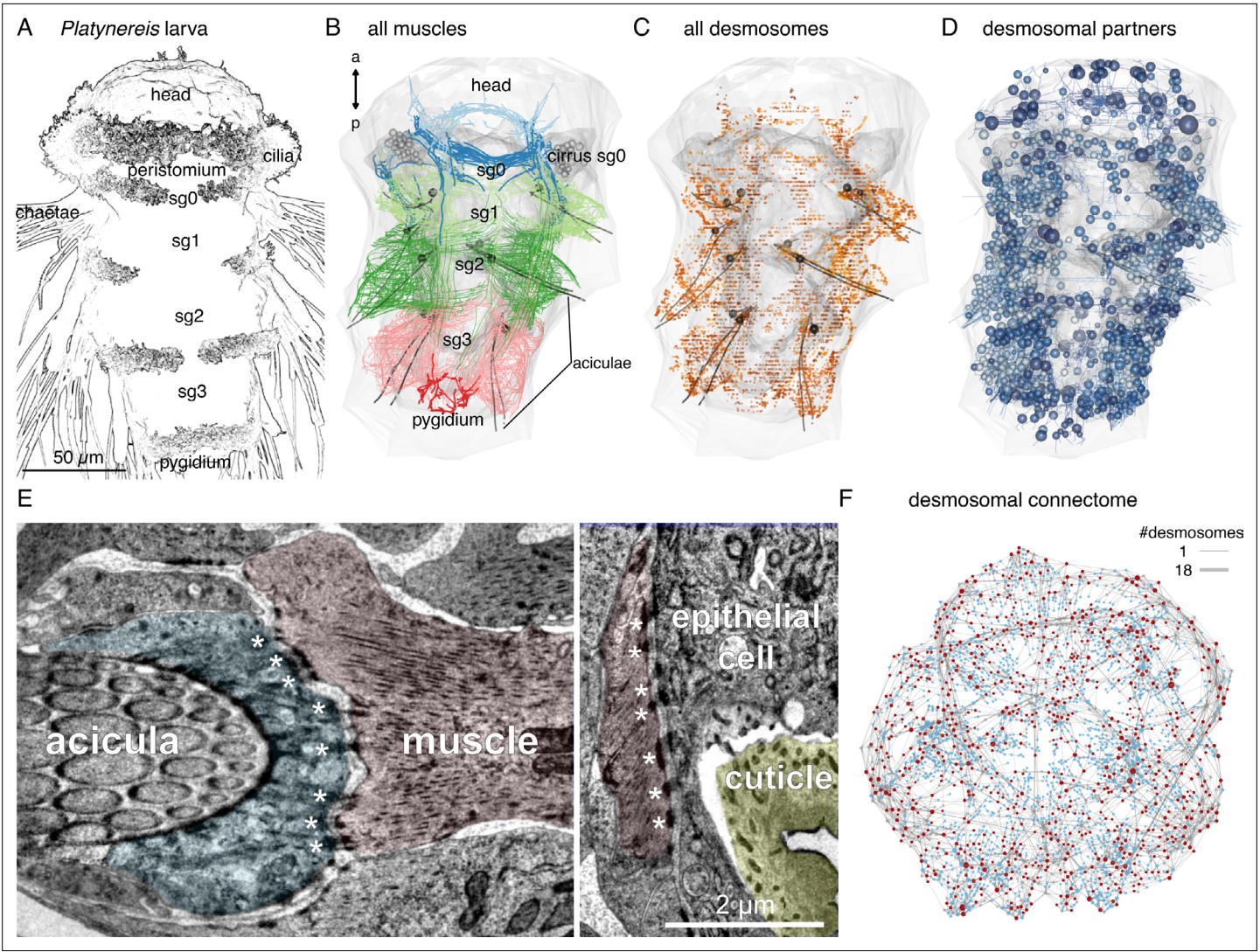

**Figure 1.** Whole-body desmosomal connectomics in the *Platynereis* larva. (**A**) Stylised scanning electron micrograph of a 3-day-old *Platynereis* larva with the main body regions indicated. (**B**) Skeleton representation of the aciculae and all muscle cells in the larva, coloured by body segment. The position and size of the cell body of the aciculoblasts are shown as grey spheres. The developing tentacular cirri of the cryptic segment (segment 0) are also shown in grey for reference. Nucleus positions are not shown for the muscle cells. (**C**) Position of all annotated desmosomes and hemidesmosomes in the volume. Desmosomes and hemidesmosomes were annotated every 50 layers in the first round of annotations, hence the appearance of lines. (**D**) Skeletonised representations of all the cells that connect to muscle cells through desmosomes. Spheres represent position and size of cell nuclei. (**E**) Transmission electron micrographs showing examples of desmosomal muscle-attachment sites (indicated with asterisks) between muscles and an aciculoblast (left panel) or an epidermal cell (right panel). (**F**) Overview of the desmosomal connectome, comprising muscle cells and their desmosomal partners. Nodes represent single cells or basal lamina fragments (red, muscles; cyan, other cells; grey, basal lamina fragments), connections represent desmosomes and hemidesmosomes. Node size is proportional to the weighted degree of nodes. Edge thickness is proportional to the number of desmosomes. The graph is undirected. Panels A–D show ventral views. In B–D, the body outline and yolk outline are shown in grey. Abbreviations: sg0–3, segments 0–3. This figure can be reproduced by running the code/Figure1.R script from the Jasek_et_al GitHub repository (*Jasek, 2022*).

The online version of this article includes the following figure supplement(s) for figure 1:

**Figure supplement 1.** Desmosome and muscle ultrastructure.

**Figure supplement 2.** Cells containing desmosomes, tonofibrils, or both.

**Figure supplement 3.** Morphological renderings of non-muscle components of the desmosomal connectome.

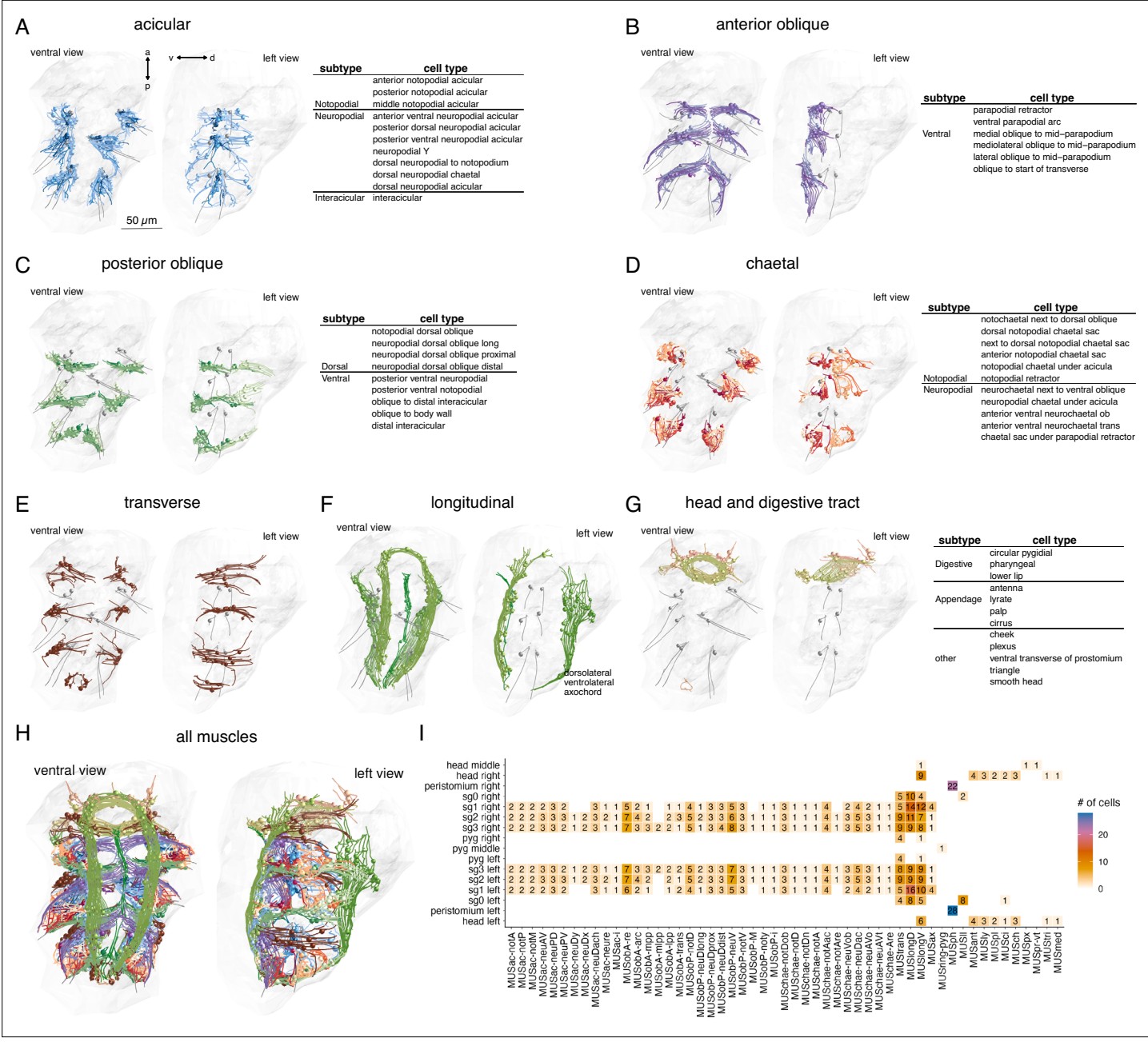

**Figure 2.** Classification and distribution of muscle cell types in the 3-day-old *Platynereis* larva. (**A**) Morphological rendering of traced skeletons and soma position (spheres) of all acicular muscles. The table shows the cell-type classification of acicular muscles. (**B**) All anterior oblique muscles and their classification. (**C**) All posterior oblique muscles and their classification. (**D**) All chaetal muscles and their classification. (**E**) All transverse muscles (one type only). (**F**) All longitudinal muscles and their classification. (**G**) Muscles of the developing digestive system and the head and their classification. (**H**) All muscles, coloured by category. (**I**) Segmental and left–right distribution of all muscle cell types. Numbers indicate the number of cells per type and body region. The short names of the individual muscle types are based on composites of anatomical abbreviations (cf. panels A–G). Not, notopodial; neu, neuropodial; A, anterior; P, posterior; D, dorsal; V, ventral; ac, acicular; chae, chaetal; ob, oblique; trans, transverse; long, longitudinal. Full muscle names and their abbreviations are listed in *Table 1*. In A–H, the left panels are ventral views, the right panels lateral-left views. The body outline is shown in grey, aciculae are shown for segmental reference. This figure can be reproduced by running the code/Figure2.R script (*Jasek, 2022*).

The online version of this article includes the following source data for figure 2:

**Source data 1.** Source data for panel I listing the segmental and left–right distribution of all muscle cell types.

**Source data 2.** All muscle skeletons as point clouds with tangent vectors (dotprop).

**Table 1.** Muscle cell types.

Hierarchical classification and nomenclature of all muscle cells in the 3-day-old *Platynereis* larva.

| | | Anatomical name and CATMAID annotation | Example CATMAID neuron name (example) | Number of cells (per segment [sg] and body side) | Desmosomal connections to | Synaptic connections from (>3 synapses) |
|---|---|---|---|---|---|---|
| Acicular muscle (MUSac) | | | | | | |
| | Notopodial | Anterior notopodial acicular muscle | MUSac-notA | sg1: 2 l, 2 r sg2: 2 l, 2 r sg3: 2 l, 2 r | Proximal base of notopodial acicula; notopodial ECs | MNring, MNcrab, MNwave, MNantacic |
| | | Posterior notopodial acicular muscle | MUSac-notP | sg1: 2 l, 2 r sg2: 2 l, 2 r sg3: 2 l, 2 r | Proximal base of notopodial acicula; septal ECs and dorsal paratroch | MNacic, MNwave, MNbow, MNspider-ant, MNcrab, MNperif |
| | | Middle notopodial acicular muscle | MUSac-notM | sg1: 2 l, 2 r sg2: 2 l, 2 r sg3: 2 l, 2 r | Proximal base of notopodial acicula; parapodial ECs | MNwave, MNacic, MNantaci, MNbow |
| | | Unpaired notopodial dorsal acicular muscle | MUSac-notDo | sg1: 0 l, 0 r sg2: 1 l, 0 r sg3: 0 l, 0 r | Proximal base of notopodial acicula; dorsal ECs | None |
| | | Anterior ventral neuropodial acicular muscle | MUSac-neuAV | sg1: 2 l, 2 r sg2: 3 l, 3 r sg3: 3 l, 3 r | Proximal base of neuropodial acicula; epidermal cells near ventrolateral muscles | MNcrab, MNring, MNchae |
| | | Posterior dorsal neuropodial acicular muscle | MUSac-neuPD | sg1: 3 l, 3 r sg2: 3 l, 3 r sg3: 3 l, 3 r | Proximal base of neuropodial acicula; dorsal ECs and notopodial EC chaeFC and distal notopodial acicula (1) and dorsal paratroch (2) | MNarm, sparse various others |
| | | Posterior ventral neuropodial acicular muscle | MUSac-neuPV | sg1: 2 l, 2 r sg2: 3 l, 3 r sg3: 2 l, 2 r | Proximal base of neuropodial acicula; ventral paratroch and ECs around it | MNcrab, MNacicX, MNperifac, MNspider-post, MNspider-ant, MNpostv |
| | | Neuropodial Y | MUSac-neuDy | sg1: 0 l, 0 r sg2: 1 l, 1 r sg3: 1 l, 1 r | Proximal and mid neuropodial acFC; neuropodial chaeFC-hemi; mid-parapodial septal EC | Fragments |
| | | Dorsal neuropodial muscle to notopodium | MUSac-neuDx | sg1: 0 l, 0 r sg2: 2 l, 2 r sg3: 2 l, 2 r | Proximal neuropodial acFC; mid-distal notopodial acFC; and mid-dorsal parapodial ECs (2) | MNchae, MNcrab, MNarm |
| | | Dorsal neuropodial chaetal muscle | MUSac-neuDach | sg1: 3 l, 3 r sg2: 3 l, 3 r sg3: 3 l, 3 r | Entire length of neuropodial acicula (all acFC); mid-parapodial ECs | MNarm, MNac, MNchae, MNantacic |
| | Neuropodial | Chaetal sac retractor | MUSac-neure | sg1: 1 l, 1 r sg2: 2 l, 2 r sg3: 1 l, 1 r | Proximal base of neuropodial acicula; proximal neuropodial chaeFC | Sparse |
| | Inter-acicular | interacicular_muscle | MUSac-i | sg1: 1 l, 1 r sg2: 1 l, 1 r sg3: 1 l, 1 r | Proximal base of neuropodial acicula; proximal base of notopodial acicula | None |

*Table 1 continued on next page*

*Table 1 continued*

| | | Anatomical name and CATMAID annotation | Example CATMAID neuron name (example) | Number of cells (per segment [sg] and body side) | Desmosomal connections to | Synaptic connections from (>3 synapses) |
|---|---|---|---|---|---|---|
| Anterior oblique muscle | Anterior ventral oblique muscles | Parapodial retractor muscle | MUSobA-re | sg1: 6 l, 5 r sg2: 7 l, 7 r sg3: 7 l, 7 r | Neuropodial EC chaeFC and chaetal sac ECs; midline cells | MNspider-ant, MNantacic, MNhose, MNob-contra |
| | | Ventral parapodial muscle arc | MUSobA-arc | sg1: 2 l, 2 r sg2: 4 l, 4 r sg3: 3 l, 3 r | Basal lamina next to nerve chord; neuropodial ECs and chaeFC-ECs | MNspider-ant, MNantacic, MNob-contra |
| | | Medial oblique to mid-parapodium | MUSobA-mpp | sg1: 1 l, 1 r sg2: 2 l, 2 r sg3: 3 l, 3 r | Medial basal lamina next to medial nerve cord and putative radial glia; mid-parapodial ECs | MNob-contra, MNhose, MNcrab, others |
| | | Mediolateral oblique to mid-parapodium | MUSobA-mlpp | sg1: 0 l, 0 r sg2: 2 l, 2 r sg3: 0 l, 0 r | Medial basal lamina next to mediolateral nerve cord and putative radial glia; mid-parapodial ECs | MNhose |
| | | Lateral oblique to mid-parapodium | MUSobA-lpp | sg1: 1 l, 1 r sg2: 2 l, 2 r sg3: 2 l, 2 r | Lateral basal lamina next to the nerve cord and putative radial glia; mid-parapodial ECs | MNob-contra, MNhose, MNsmile, other |
| | | Oblique to start of transverse | MUSobA-trans | sg1: 1 l, 1 r sg2: 2 l, 2 r sg3: 2 l, 2 r | Basal lamina next to the axochord; ventrolateral ECs | MNhose, MNcross |

*Table 1 continued*

| Anatomical name and CATMAID annotation | | | Example CATMAID neuron name (example) | Number of cells (per segment [sg] and body side) | Desmosomal connections to | Synaptic connections from (>3 synapses) |
|---|---|---|---|---|---|---|
| | Posterior dorsal oblique muscle | Notopodial dorsal oblique muscle | MUSobP-notD | sg1: 4 l, 4 r sg2: 5 l, 5 r sg3: 5 l, 5 r | Dorsal longitudinal muscles and ECs near them; ECs of the notopodium | MNpostacic, MNspider-ant |
| | | Neuropodial dorsal oblique long | MUSobP-neuDlong | sg1: 1 l, 1 r sg2: 2 l, 2 r sg3: 2 l, 1 r | MUSlong_D; distal neuropodial acFC, neuropodial ECs, and mid-parapodial ECs | MNspider-post; MNpostacic |
| | | Neuropodial dorsal oblique proximal | MUSobP-neuDprox | sg1: 3 l, 3 r sg2: 3 l, 3 r sg3: 3 l, 3 r | MUSlong_D; dorsal paratroch and ECs around it | MNspider-post |
| | | Neuropodial dorsal oblique distal | MUSobP-neuDdist | sg1: 3 l, 3 r sg2: 3 l, 3 r sg3: 3 l, 3 r | ECs near dorsal paratroch; distal neuropodial acFC, neuropodial ECs and mid-parapodial ECs | MNbow |
| Posterior oblique muscle | Posterior ventral oblique muscle | Posterior ventral neuropodial muscle | MUSobP-neuV | sg1: 5 l, 5 r sg2: 7 l, 7 r sg3: 7 l, 7 r | Basal lamina next to VNC; ECs from the distal part of neuropodial acicula to the ventral paratroch area | MNring, MNspider-post, MNob-ipsi, MN_oblique, MNob, many fragments |
| | | Posterior ventral notopodial muscle | MUSobP-notV | sg1: 3 l, 3 r sg2: 3 l, 3 r sg3: 3 l, 3 r | Basal lamina next to VNC; septal ECs; EC and chaeFC near distal part of notopodial acicula | MNhose, MNspider-post, MNob-ipsi, MN_oblique, MNladder, MNob-contra, fragments |
| | | Oblique to distal inter-acicular | MUSobP-M | sg1: 0 l, 0 r; sg2: 1 l, 1 r sg3: 1 l, 1 r | Each other and basal lamina of the VNC; distal inter-acicular muscle | None |
| | | Oblique to body wall near distal inter-acicular and neuropodial Y | MUSobP-noty | sg1: 1 l, 1 r sg2: 1 l, 1 r sg3: 1 l, 1 r | Basal lamina next to the axochord; septal ECs | MNhose, MNpostacic, MNob-ipsi, MNspider-post, fragments |
| | | sg0 ventral posterior oblique muscle | MUSobP | sg0: 2 l, 2 r sg1: 0 l, 0 r sg2: 0 l, 0 r sg3: 0 l, 0 r | Basal lamina near VNC and ventral ECs; dorsal ECs | MNsmile, MNcrab, MNob-contra, MNmouth |
| | Posterior median oblique muscle | Distal inter-acicular muscle | MUSobP-i | sg1: 1 l, 1 r sg2: 1 l, 1 r sg3: 1 l, 1 r | Neuropodial EC, chaeFC-EC, acFC-EC; notopodial EC, chaeFC EC; MUSob-postM | Sparse |
| | | Oblique muscle other | MUSob | sg0: 1 l, 1 r sg1: 0 l, 0 r sg2: 0 l, 0 r sg3: 0 l, 0 r | ECs near mouth/lower lip; MUStrans | MNsmile |

*Table 1 continued on next page*

Table 1 continued

| Anatomical name and CATMAID annotation | | Example CATMAID neuron name (example) | Number of cells (per segment [sg] and body side) | Desmosomal connections to | Synaptic connections from (>3 synapses) |
|---|---|---|---|---|---|
| | | Notochaetal next to dorsal oblique | MUSchae-notDob | sg1: 3 l, 3 r sg2: 3 l, 3 r sg3: 3 l, 3 r | Dorsolateral ECs; notopodial chaeFC | MNpostatic, various |
| | | Dorsal notopodial chaetal sac muscle | MUSchae-notD | sg1: 1 l, 1 r sg2: 1 l, 1 r sg3: 1 l, 1 r | Proximal notopodial acicula; notopodial chaeFC (semicircle around dorsal side of notopodial chaetal sac) | Sparse |
| | | Next to dorsal notopodial chaetal sac muscle | MUSchae-notDn | sg1: 1 l, 1 r sg2: 1 l, 1 r sg3: 1 l, 1 r | Dorsolateral ECs; ECs of the notopodial chaetal sac and notopodial chaeFC-ECs; notopodial acFC cells | MNpostacic |
| | | Anterior notopodial chaetal sac muscle | MUSchae-notA | sg1: 1 l, 1 r sg2: 1 l, 1 r sg3: 1 l, 1 r | Mid-parapodial chaetal sac ECs and MUSobA-che; notopodial ECs, chaeFC-EC and acFC | MNspider-ant, MNantacic |
| | Notopodial | Notopodial chaetal muscle under acicula | MUSchae-notAac | sg1: 4 l, 4 r sg2: 4 l, 4 r sg3: 4 l, 4 r | Distal acFC; all types of chaetal sac cells | MNcrab, MNbiramous, MNwave, MNantacic, MNacic, MNspider-ant, MNbow, fragments |
| | | Notopodial retractor muscle | MUSchae-notAre | sg1: 0 l, 0 r sg2: 1 l, 1 r sg3: 1 l, 1 r | Mid-parapodial ECs; distal notopodial acFC and EC chaeFC | Sparse |
| | | Neurochaetal next to ventral oblique | MUSchae-neuVob | sg1: 2 l, 2 r sg2: 3 l, 3 r sg3: 2 l, 2 r | Proximal neuropodial chaeFC-hemi; ventrolateral ECs | MNchae |
| | | Neuropodial chaetal muscle under acicula | MUSchae-neuDac | sg1: 4 l, 4 r sg2: 5 l, 5 r sg3: 5 l, 5 r | Mid and distal neuropodial circumacicluar cells; proximal and distal chaeFC cells and chaetal sac ECs | MNbiramous, MNchae, MNspider-ant, fragments |
| | | Anterior ventral neurochaetal muscle ob | MUSchae-neuAVo | sg1: 2 l, 2 r sg2: 3 l, 3 r sg3: 3 l, 3 r | Neuropodial mid acFC and proximal chaeFC; distal chaeFC and neuropodial ECs | MNarm, MNchae |
| | | Anterior ventral neurochaetal muscle trans | MUSchae-neuAVt | sg1: 1 l, 1 r sg2: 1 l, 1 r sg3: 1 l, 1 r | Distal neuropodial acFC; ventrolateral ECs next to VLM | MNacicX, MNspider-post |
| Chaetal sac muscle | Neuropodial | Chaetal sac under parapodial retractor | MUSchae-Are | sg1: 1 l, 1 r sg2: 1 l, 1 r sg3: 1 l, 1 r | Mid and distal neuropodial chaeFC cells | MNbiramous, MNspider-ant |
| Transverse muscle | | Transverse muscle | MUStrans | sg0: 4 l, 5 r sg1: 5 l, 5 r sg2: 9 l, 9 r sg3: 8 l, 9 r pyg: 4 l, 4 r | Lateral ECs; metatroch; akrotroch | MNring, MNhose, MNsmile, MNob-contra, MNladder |

Table 1 continued on next page

Table 1 continued

| | Anatomical name and CATMAID annotation | Example CATMAID neuron name (example) | Number of cells (per segment [sg] and body side) | Desmosomal connections to | Synaptic connections from (>3 synapses) |
|---|---|---|---|---|---|
| Longitudinal muscle | Dorsolateral | Dorsolateral muscle | MUSlongD | sg0: 11 l, 15 r sg1: 13 l, 15 r sg2: 10 l, 10 r sg3: 7 l, 7 r | Epidermal cells; MUSobP-neuD, MUSobP-notDprox, MUSobP-notDlong; paratroch; nuchal organ | MN2, MNring, MNcrab |
| | Ventrolateral | Ventrolateral muscle | MUSlongV | head: 9 l, 8 r, 1 m sg0: 7 l, 4 r sg1: 8 l, 12 r sg2: 11 l, 8 r sg3: 7 l, 7 r pyg: 1 l, 1 r | Basal lamina lateral of VNC; ventral ECs | MN1, MNring, MNcrab, MNsmile, MNspider-ant |
| | Axochord | Axochord | MUSax | sg1: 4 l, 4 r sg2: 1 l, 1 r sg3: 1 l, 1 r | Basal lamina next to VNC; radial glia-like midline; ECs | MNax; MNsmile |
| Digestive system muscle | Circular pygidial muscle | | MUSring-pyg | pyg1 | Self; dorsal pygidial ECs; ventral pygidial ECs | None |
| | Pharyngeal muscle | | MUSph | 28 l, 30 r | Basal lamina | None |
| | Lower lip muscle | | MUSll | 5 l, 5 r | Metatroch and ECs next to metatroch | MNsmile |
| Head muscle | Head appendage | Antenna muscle | MUSant | 4 l, 4 r | ECs | Sparse |
| | | Lyrate muscle | MUSly | 3 l, 3 r | Basal lamina | None |
| | | Palp muscle | MUSpl | 2 l, 2 r | ECs and prototroch cover cells | None |
| | | Cirrus muscle | MUSci | 2 l, 2 r | ECs and basal lamina | None |
| | | Cheek muscle | MUSch | 3 l, 3 r | Basal lamina | MNsmile |
| | | Plexus muscle | MUSpx | 1 | ECs and putative glia | SNs with very weak connections |
| | | Ventral transverse muscle of prostomium | MUSpr-Vt | 1 | Dorsolateral head ECs | Sparse |
| | | Triangle muscle | MUStri | 1 l, 1 r | Basal lamina | None |
| | Other head | Smooth head muscles | MUSmed-head | 1 l, 1 r | Head ECs | None |

All anatomical reconstructions are available at https://catmaid.jekelylab.ex.ac.uk or can be accessed by the R code provided. For the 853 muscle cells, we also provide the skeletons as point clouds with tangent vectors (dotprops) (*Figure 2—source data 2*).

## Classification, left–right stereotypy, and segmental distribution of muscle cell types

The eight main muscle groups could be further subdivided into 53 distinct types (*Figure 2*, *Table 1*, *Video 1*). These 53 types represent anatomically distinct groups of cells that we classified based on their position, shape, motoneuron inputs, and desmosomal partners. All types are composed of segmentally repeated sets with the same number of cells on the left and right body sides, with few exceptions (*Figure 2I*; *Table 1*). The precise and repeated features of the anatomy and the type-specific patterns of desmosomal connectivity and motoneuron innervation suggest that these 53 types represent cell types specified by unique development programmes and muscle identity genes, as is the case for example in *Drosophila* larvae (*de Joussineau et al., 2012*).

To name each muscle cell type we adapted available names from the annelid anatomical literature (*Allentoft-Larsen et al., 2021*; *Bergter et al., 2008*; *Filippova et al., 2010*; *Mettam, 1967*). However,

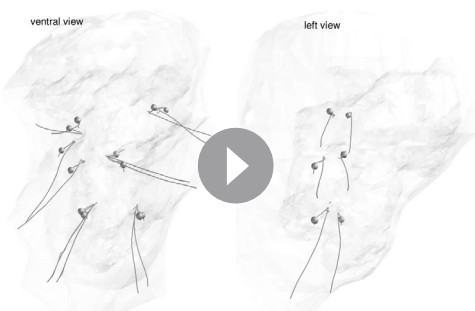

**Video 1.** All muscles in the 3-day-old *Platynereis* larva. Individual muscle cell types are shown by cell type and muscle category. The 12 aciculae are shown in grey for reference. The cell nuclei are labelled by a sphere. The yolk and body outlines are shown in grey. This video can be reproduced by loading the Jasek_et_al.Rproj R project in RStudio and running the code/Video1.R script (*Jasek, 2022*).

https://elifesciences.org/articles/71231/figures#video1

## desmosomal connectome

To analyse how muscles connect to each other and to the rest of the body, we focused on desmosomes. We use a morphological definition of desmosomes as electron-dense structures linking two

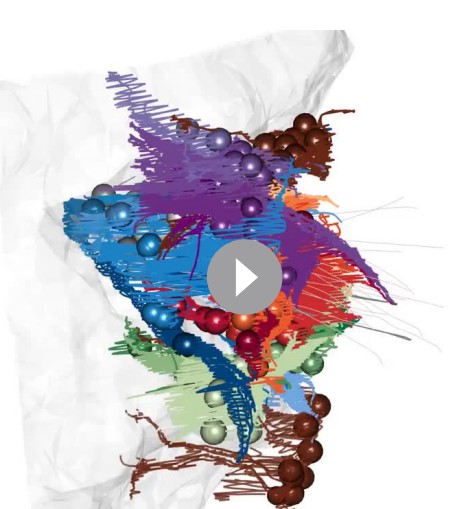

**Video 2.** Muscle groups in the parapodial complex. Reconstruction of all cell groups in the left parapodial complex in the second segment. The neuropodial and notopodial aciculae are shown in black, chaetae are in yellow. The dark-brown cell with the large nucleus is the spinning gland. The yolk outline is shown in grey. Subsequent rotations highlight selected muscle cell types involved in some of the movements in Figure 6—figure supplement 4. CATMAID view of a similar 3D rendering: https://catmaid.jekelylab.ex.ac.uk/11/links/ooeymw3. This video can be reproduced by loading the Jasek_et_al.Rproj R project in RStudio and running the code/Video2.R script (*Jasek, 2022*).

https://elifesciences.org/articles/71231/figures#video2

our study is the highest resolution whole-body analysis to date, therefore we also had to extend and modify the available nomenclature (*Figure 2* and *Table 1*).

The majority of muscle types (34 out of 53 types corresponding to 502 out of 853 cells) are in the parapodial appendages and form the parapodial muscle complex (*Video 2*). The parapodial complex of the first segment in the 3-day-old larva has the same major muscle groups as other segments, but fewer muscle cells and some minor subgroups are missing (MUSac-neuDy, MUSac-neuDx, and MUSobP-M). The parapodia of this segment will transform into tentacular cirri during cephalic metamorphosis (*Fischer et al., 2010*), but at the 3-day-old stage these are bona fide locomotor appendages. The 3-day-old larva also has a cryptic segment (segment 0) (*Steinmetz et al., 2011*), which lacks parapodia and has few muscle cells (*Figure 2I*, *Table 1*).

## Reconstruction of the whole-body

adjacent cells (desmosomes) or cells to the basal lamina (hemidesmosomes). In non-muscle cells, desmosomes are often anchored by cytoplasmic tonofibrils, thick bundles of intermediate filaments (*Figure 1—figure supplement 1*, *Figure 1—figure supplement 2*). In the volume, we sampled and annotated desmosomes and hemidesmosomes and defined the partner cells (or basal lamina fragments) that were interconnected by them. We used this connectivity information and the cell annotations to derive a 'desmosomal connectome' containing muscle and other cells and basal lamina fragments (short, locally-traced skeletons) as nodes (vertices) and desmosomes and hemidesmosomes as links (edges). The desmosomal connectome forms a single large interconnected network (*Figure 1*, *Figure 3*) after the exclusion of a few small isolated subgraphs (see Methods).

Desmosomes are one of the strongest types of adhesive junction, characteristic of tissues that experience mechanical stress. The position and density of desmosomes in muscle cells in the *Platynereis* larva suggest that these are the primary junctions mediating force transmission. Other types of junctions including various types of adherens-like junctions (*Tilic and Bartolomaeus, 2016*) (primarily found in epidermal and chaetal follicle cells) were not considered here. Some adjacent muscle cells are also connected by

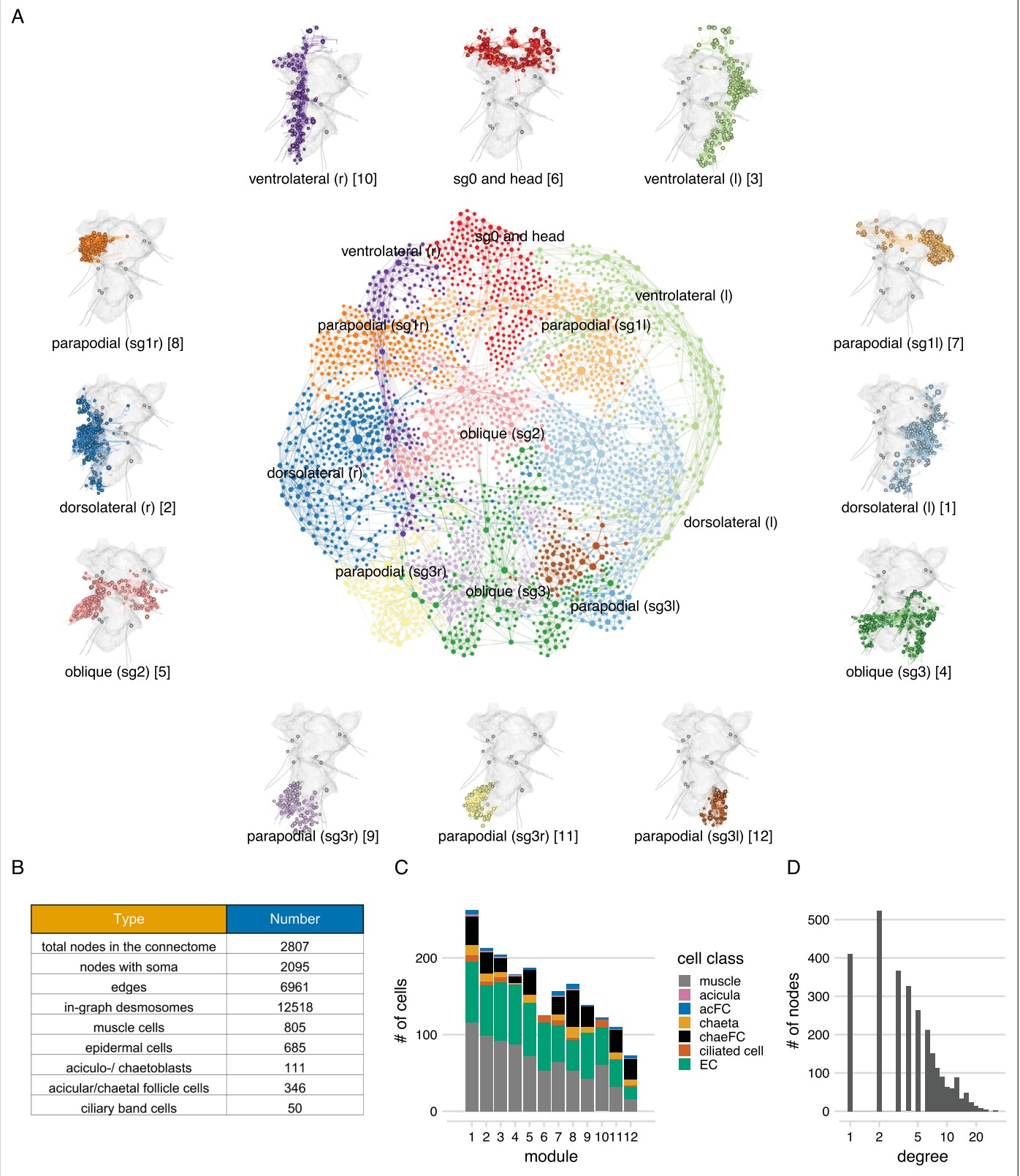

**Figure 3.** The desmosome connectome of the 3-day-old *Platynereis* larva. (**A**) The desmosomal connectome coloured by Leiden modules. Nodes represent cells or basal lamina fragments and edges represent desmosomal connections. Node sizes are proportional to weighted degree (sum of all weighted connections). The graph layout was computed by a force-field-based method. Around the network graph, morphological renderings of the cells are shown for each module. Spheres show positions and sizes of nuclei. Grey meshes show the outline of the yolk. The 12 aciculae are shown for

*Figure 3 continued on next page*

*Figure 3 continued*

segmental reference. Numbers in square brackets after the module names show module id, ordered by module size. (**B**) Number of nodes, edges, and cells per main cell classes in the desmosomal connectome. (**C**) Number of cells in each module (ordered by module size) coloured by cell class. (**D**) Histogram of node degrees (number of connected nodes) for the desmosomal connectome. This figure can be reproduced by running the code/Figure3.R script (*Jasek, 2022*).

The online version of this article includes the following source data and figure supplement(s) for figure 3:

**Source data 1.** The desmosomal connectome graph in igraph format.

**Source data 2.** The desmosomal connectome graph in visNetwork format.

**Source data 3.** The desmosomal connectome graph in html format.

**Figure supplement 1.** Grouped connectivity graph of the desmosomal connectome.

**Figure supplement 1—source data 1.** The grouped desmosomal connectome graph in visNetwork format.

**Figure supplement 1—source data 2.** The grouped desmosomal connectome graph in html format.

an unclassified type of adhaerens-like junction (some examples are tagged with 'junction type 3' in the volume). These junctions are not in every muscle, are less electron-dense and were not annotated systematically.

We first annotated (hemi-)desmosomes in every 50 layers of the 4847 layer EM dataset. We then manually surveyed each muscle cell and identified their desmosomal partners that were not connected in the first survey. Desmosomes can span multiple 40 nm EM layers (up to 15 layers) and are enriched at the ends of muscle cells indicating that they transmit force upon muscle cell contraction (*Figure 1E*, *Figure 1—figure supplement 1*). The (hemi-)desmosomal partners of muscle cells include the basal lamina (34.4% of desmosomes) and a diversity of cell types. These are other muscle cells, glia, multiciliated cells of the ciliary bands (except the prototroch), epidermal cells, and various follicle cells encircling the chaetae (chaetal follicle cells) and the aciculae (acicular follicle cells) (*Figure 1—figure supplement 2A*, *Figure 1—figure supplement 3*). Only 2.4% of (hemi-)desmosomes are between muscle cells and 60.5% of these are between different muscle types. This suggests that in the *Platynereis* larva, desmosomes do not mediate lateral force transmission between muscles of the same bundle. In many desmosomal partner cell types – but not in the muscle cells themselves – we could identify tonofibrils (*Figure 1—figure supplement 1* and *Figure 1—figure supplement 2*).

The full desmosomal connectome is an undirected graph of 2807 interconnected nodes (2095 with a soma) connected by 6961 edges. Six hundred and thirty nodes are fragments of the basal lamina of similar skeleton sizes (see Methods).

## Local connectivity and modular structure of the desmosomal connectome

To characterise the structure of the desmosomal network, we first analysed its modularity by the Leiden algorithm, which partitions graphs into well-connected communities (*Traag et al., 2019*). We detected several dense clusters of nodes or

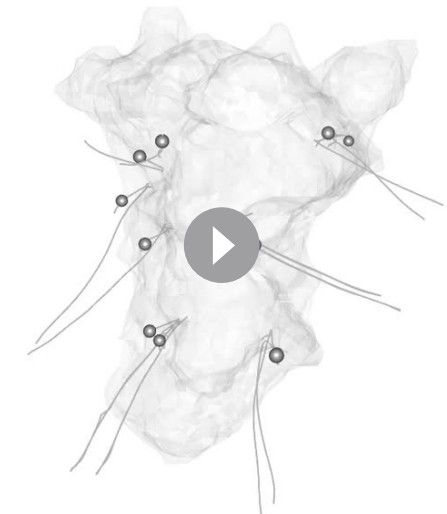

**Video 3.** 3D visualisation of all cells in the desmosomal connectome coloured by module. The colour scheme is the same as in Figure 3. The cell nuclei are labelled by a sphere. Individual Z planes from the stack are also shown to indicate the orientation of the serial EM stack. The yolk and body outlines are shown in grey. CATMAID view: https://catmaid.jekelylab.ex.ac.uk/11/links/7ftc5sa. This video can be reproduced by loading the Jasek_et_al.Rproj R project in RStudio and running the code/Video3.R script (*Jasek, 2022*).
https://elifesciences.org/articles/71231/figures#video3

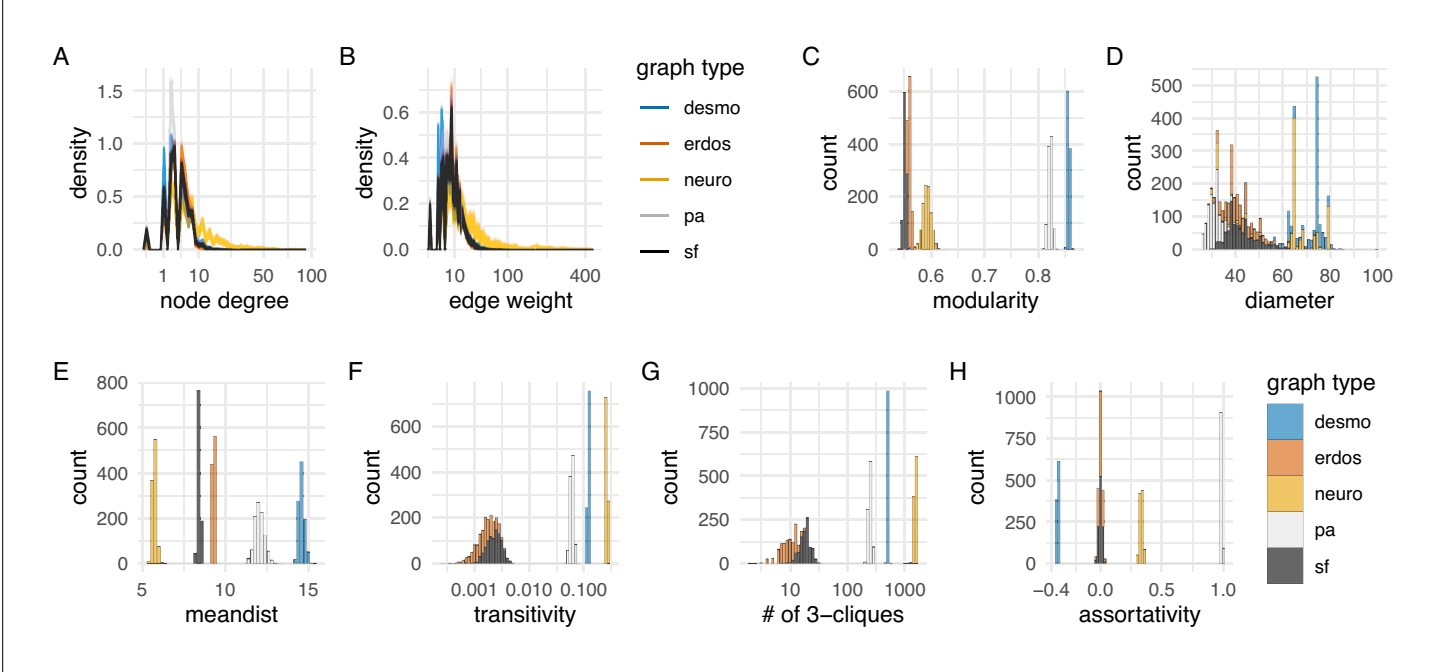

**Figure 4.** Network statistics of the desmosomal connectome relative to random graphs. (**A**) Degree distribution and (**B**) edge-weight distribution of the desmosomal connectome compared to the synaptic (neuronal) connectome (neuro), scale-free (sf), Erdős-Rényi (erdos), and preferential-attachment (pa) graphs (1000 graphs each). (**C**) Modularity scores of 1000 weighted scale-free (sf), Erdős-Rényi, and preferential-attachment (pa) graphs relative to the weighted desmosomal and synaptic connectome graphs (1000 subsamples each). (**D**) Mean diameter scores of 1000 scale-free, Erdős-Rényi, and preferential-attachment graphs relative to the desmosomal and synaptic connectome graphs (1000 subsamples). (**E**) Mean distance and (**F**) transitivity (clustering coefficient) scores of 1000 weighted scale-free, Erdős-Rényi, and preferential-attachment graphs relative to the weighted desmosomal and synaptic connectome graphs (1000 subsamples each). (**G**) Number of 3-member cliques (triangles) and (**H**) assortativity coefficient. *X* axes are in sqrt scale for A, B and log10 scale for F, G. Abbreviations: desmo, desmosomal connectome; neuro, synaptic connectome; sf, scale-free; erdos, Erdős-Rényi; pa, preferential-attachment. This figure can be reproduced by running the code/Figure4.R script (*Jasek, 2022*).

The online version of this article includes the following source data for figure 4:

**Source data 1.** Network statistics for the 1,000 simulated scale-free, Erdős-Rényi, preferential-attachment graphs and the 1000 subsampled desmosomal connectome and synaptic connectome graphs.

communities in the desmosomal connectome. These communities correspond to anatomical territories in the larval body (*Figure 3*, *Video 3*). There are four modules – a left and right ventrolateral and a left and right dorsolateral – consisting of longitudinal muscles spanning all body segments, and associated ciliary band and epidermal cells. Various head and segment-0 muscles constitute another module (*Figure 3*, *Video 3*). Two modules contain left–right groups of parapodial and oblique muscles in the second and third segments, connected at the midline through the basal lamina and the median ventral longitudinal muscle (axochord or MUSax) (*Lauri et al., 2014*; *Purschke and Müller, 2006*). The other modules include parapodial muscles and chaetal sac cells, in the segmental parapodia. Each module contains muscles and epidermal cells, and various other cell types (*Figure 3C*).

We also generated a grouped connectivity graph where cells of the same type were collapsed into one node (*Figure 3—figure supplement 1*).

To better understand the organisation of the desmosomal network, we compared it to simulated networks generated by three different stochastic algorithms and a reduced synaptic (neuronal) connectome graph from the same *Platynereis* larva (*Veraszto et al., 2020*). We used 1000 simulated graphs for each method and 1000 subsampled graphs from the connectome graphs.

While all graph types have a similar degree and weighted degree distribution (*Figure 4A, B*), the desmosomal graph stands out as a highly modular and less compact graph with strong local connectivity. This conclusion is supported by several graph measures.

The modularity scores of the subsampled desmosomal graphs are higher than for any other graph types, including the synaptic connectome (*Figure 4C*). The desmosomal subgraphs have the largest

graph diameter (the length of the longest graph geodesic between any two vertices) and the mean of the distances between vertices (*Figure 4D, E*), both measures of graph compactness (*Doyle and Graver, 1977*). The desmosomal and synaptic connectome graphs have the highest transitivity values (clustering coefficient), which measures the probability that nodes connected to the same node are also directly connected to each other (*Figure 4F*). A similar measure, the number of 3-cliques (three fully connected nodes) also ranks the synaptic graph as first followed by the desmosomal graph (*Figure 4G*). The assortativity coefficient (between −1 and 1), a measure of the extent to which nodes of similar degree are connected (1 for the preferential-attachment graphs) is lowest for the desmosomal graph, indicating more connections between nodes of different degree (*Figure 4H*).

Overall, the desmosomal graph stands out as a highly modular graph with a large diameter, large average distance, and high level of local connectivity (cliques). These properties set the desmosomal graph apart from different types of random graphs and the synaptic connectome. We attribute these characteristics to the special organisation of the desmosomal connectome, with all cells only connecting to cells in their immediate neighbourhood forming local cliques, and without long-range connections (e.g. between the left and right body sides).

The tight local connectivity of the desmosomal connectome is also apparent on force-field-based layouts of the graph. In force-field-based layouts, an attraction force is applied between connected nodes, together with a node-to-node repulsion and a general gravity force. As a result, more strongly connected nodes tend to be placed closer to each other.

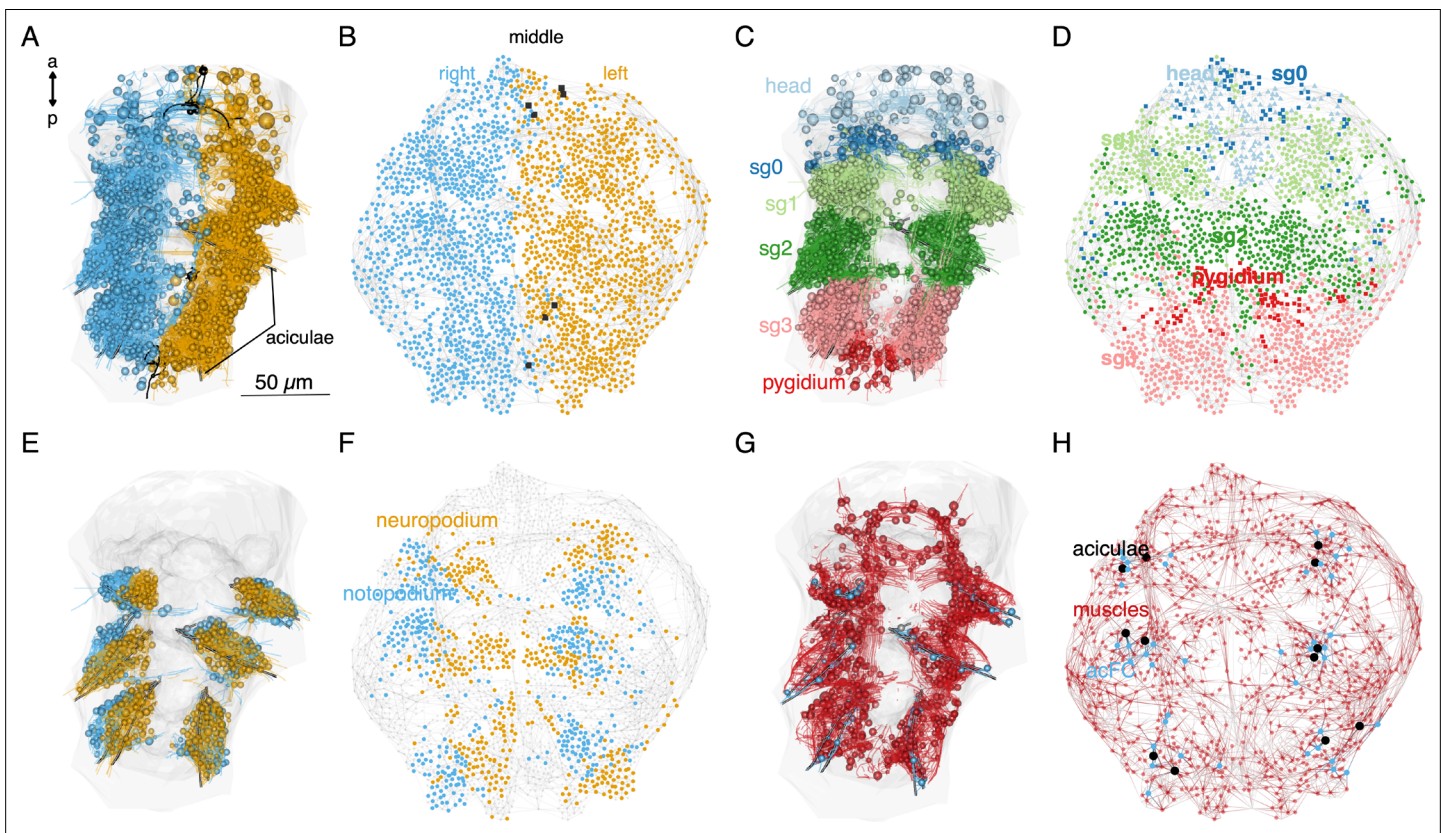

**Figure 5.** Local connectivity of the desmosomal connectome. (**A**) Morphological rendering of cells of the desmosomal connectome in the midline (black) and on the right (cyan) and left (orange) side of the body. (**B**) The same cells in the force-field-based layout of the graph, coloured by the same colour scheme. (**C**) Morphological rendering of cells of the desmosomal connectome coloured by body region (head, segments 0–3 and pygidium). (**D**) The same cells in the force-field-based layout of the graph, coloured by the same colour scheme. (**E**) Morphological rendering of cells of the desmosomal connectome in the neuropodia (orange) and notopodia (cyan). (**F**) The same cells in the force-field-based layout of the graph, coloured by the same colour scheme. (**G**) Morphological rendering of muscle cells (red), aciculae (black), and acicular follicle (acFC) cells (cyan). (**H**) The same cells in the force-field-based layout of the graph, coloured by the same colour scheme. Spheres in the morphological rendering represent the position and size of cell nuclei. Nodes in the connectivity graphs represent cells with their sizes proportional to weighted degree. This figure can be reproduced by running the code/Figure5.R script (*Jasek, 2022*). An interactive html version of the graph with node labels is in the GitHub repository supplements/Fig5_desmo_connectome_seg_interactive.html.

To analyse how closely the force-field-based layout of the desmosomal connectome reflects anatomy, we coloured the nodes in the graph based on body regions (*Figure 5*). In the force-field layout, nodes are segregated by body side and body segment. Exceptions include the dorsolateral longitudinal muscles (MUSlongD) in segment-0. These cells connect to dorsal epidermal cells that also form desmosomes with segment-1 and segment-2 MUSlongD cells. These connections pull the MUSlongD_sg0 cells down to segment-2 in the force-field layout (*Figure 5D*).

Nodes corresponding to the neuropodial (ventral) and notopodial (dorsal) parts of the parapodia also occupy distinct domains (*Figure 5E, F*). The 12 aciculae and the acicular follicle cells also occupy positions in the force-field map paralleling their anatomical positions (*Figure 5G, H*). The observation that node positions in the force-field layout of the graph recapitulate the anatomical positions of the nodes reflects the local connectedness (neighbours only) of the desmosomal connectome. This is different from the synaptic connectome where long-range neurite projections can link neurons at two different ends of the body (*Verasztó et al., 2020*).

## High muscle diversity and strong connectivity of aciculae in the parapodial complex

We further queried the desmosomal connectome to better understand the organisation and movement of the annelid larval body. We focused on the parapodia as these contain the largest diversity of muscle types and associated cells (*Video 2*).

In the segmented larva, each parapodium has a dorsal and a ventral lobe (notopodium and neuropodium). Each lobe is supported by an internal chitinous rod called acicula (*Hausen, 2005*) that is ensheathed by acicular follicle cells (*Figure 1—figure supplement 3A, C*). Each parapodial lobe also contains a chaetal sac consisting of bristle-producing chaetoblasts, chaetal follicle cells, epidermal cells (*Figure 1—figure supplement 3B, C*), and bristle mechanosensory neurons (chaeMech) (*Verasztó et al., 2020*). The chaetal follicle cells ensheathing the chaetae have four types arranged in a proximo-distal pattern (*Figure 6—figure supplement 1*). The aciculoblasts, chaetoblasts, and the various follicle cells serve as anchor points for several major parapodial muscle groups (*Figure 6E–I*, *Figure 1E*, *Video 4*).

The aciculae and associated structures in the parapodial complex represent highly connected components in the desmosomal connectome (*Figure 6E–G*). The cells with the largest mean degree and weighted degree belong to the parapodial complex. When we plot all cells in the desmosomal connectome with a colour-transparency inversely proportional to node weight, the parapodial muscle complex is highlighted (*Figure 6B, C*).

We identified 16 muscle types with connections to the aciculae or to acicular follicle cells (*Figure 6F, G*; *Figure 6—figure supplement 2*; *Figure 6—figure supplement 3*). These include acicular muscles (MUSac), some chaetal muscles (MUSchae) that link the aciculae to the chaetae and oblique muscles (MUSob) that link the aciculae to the dorsal longitudinal muscles (MUSlongD).

Several acicular muscles attach on one end to the proximal base of the aciculae and on the other end to the paratrochs and epidermal cells. Oblique muscles attach to the basal lamina, epidermal, and midline cells at their proximal end, run along the anterior edge of parapodia and attach to epidermal

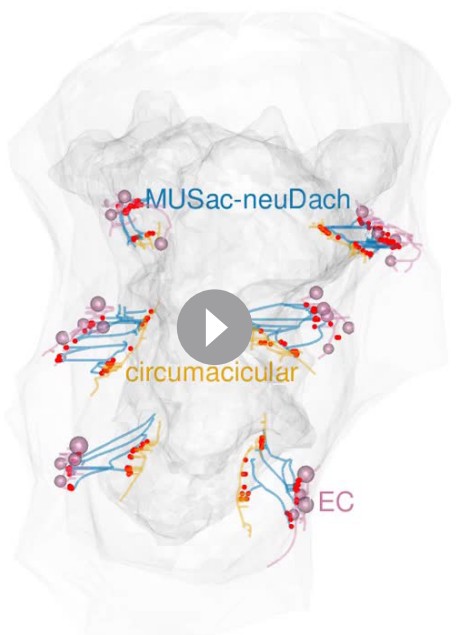

**Video 4.** 3D visualisation of selected muscle groups and their desmosomal connections. Morphological renderings of selected muscle groups and partner cells and the desmosomes (in red) that connect them. The video illustrates that desmosomes often occur at the most distal ends of muscle cells. This video can be reproduced by loading the Jasek_et_al.Rproj R project in RStudio and running the code/Video4.R script (*Jasek, 2022*).

https://elifesciences.org/articles/71231/figures#video4

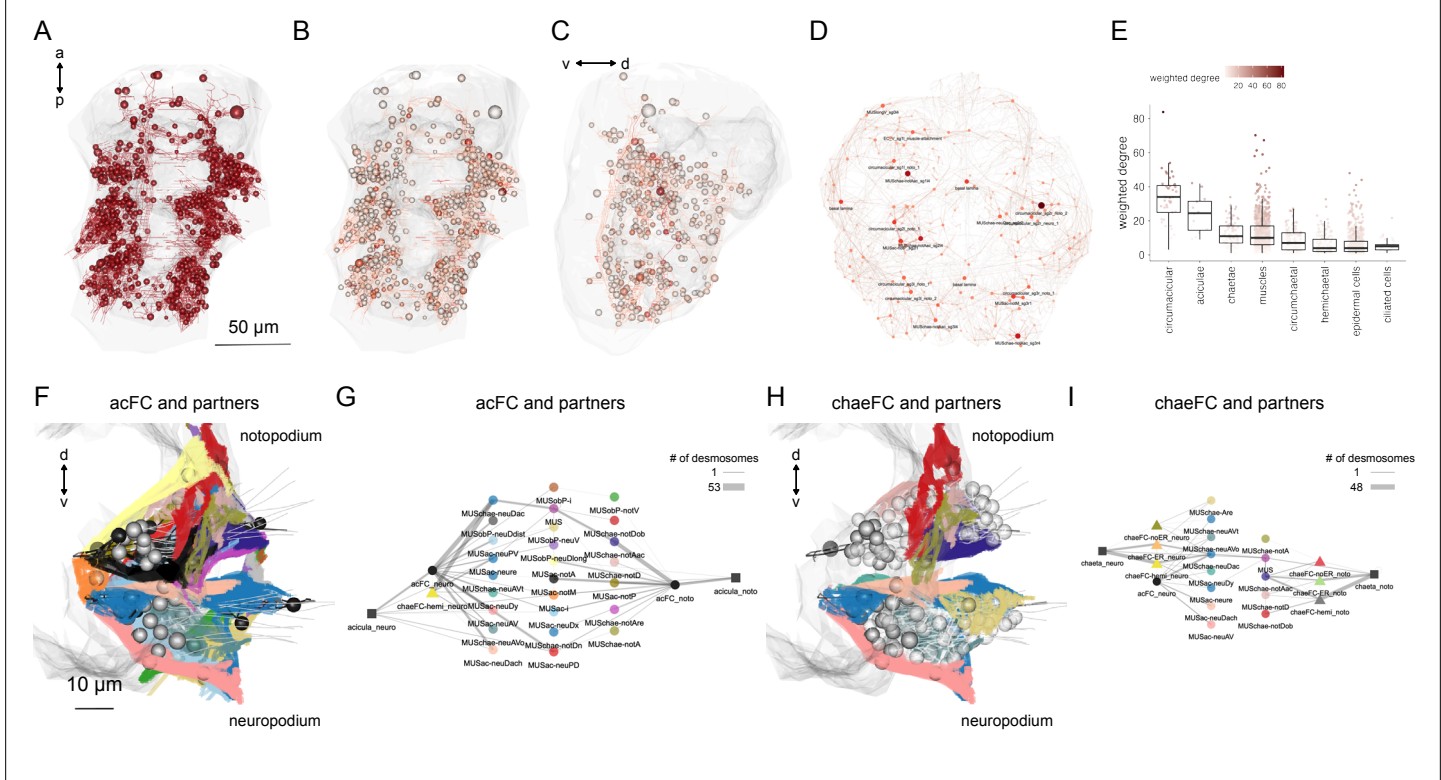

**Figure 6.** Cell-type diversity and connectivity of the parapodial muscle complex. (**A**) Morphological rendering of all cells of the desmosomal connectome with a weighted degree >10. (**B**) Same cells coloured with a colour scale proportional to the cell's weighted degree, ventral and (**C**) lateral views. (**D**) The desmosomal connectome with node-colour intensity and node size proportional to node weighted degree. (**E**) Weighted degree of the most highly connected cells in the desmosomal connectome, arranged by cell class. Colour scale in E also applies to B–D. (**F**) Morphological rendering of the outlines of muscle cell types that directly connect through desmosomes to the acicular follicle cells. Anterior view of a transverse section showing neuro- and notopodia in the left side of the second segment. (**G**) Desmosomal connectivity graph of acicular follicle cells and their partners. Nodes represent groups of cells of the same cell type. Aciculae and acicular follicle cells are separated into neuropodial and notopodial groups. (**H**) Morphological rendering of the outlines of muscle cell types that directly connect through desmosomes to the chaetal follicle cells. Anterior view of a transverse section showing neuro- and notopodia in the left side of the second segment. (**I**) Desmosomal connectivity graph of chaetal follicle cells and their partners. Nodes represent groups of cells of the same cell type. Chaetae and chaetal follicle cells are separated into neuropodial and notopodial groups. Edge thickness is proportional to the number of desmosomes connecting two cell groups. This figure can be reproduced by the code/Figure6.R script (**Jasek, 2022**).

The online version of this article includes the following source data and figure supplement(s) for figure 6:

**Source data 1.** Source data for panel E of *Figure 6*.

**Figure supplement 1.** Ultrastructure of chaetal sac cells.

**Figure supplement 2.** Morphological renderings of oblique muscles.

**Figure supplement 3.** Morphological renderings of acicular and chaetal muscles.

**Figure supplement 4.** Morphological renderings of muscles in the parapodial muscle complex and inferred movement of the aciculae.

and chaetal follicle cells at their distal tips (*Figure 6—figure supplement 2*, *Videos 2 and 4*). Both of these muscle groups are involved in moving the entire parapodium. Acicular muscles move the proximal tips of the aciculae, while oblique muscles move the parapodium by moving the tissue around the chaetae and the aciculae. All acicular movements also correspond to parapodial movements. Chaetae are embedded in the parapodium and therefore move with it, but the chaetal sac muscles can also independently retract the chaetae into the parapodium or protract them and make them fan out.

The notopodial and neuropodical aciculae have distinct muscle partners indicating that these structures could move independent of each other. Two muscles (MUSac-i and MUSac-neuDx) link the noto- and neuropodial aciculae within the same parapodium to each other, suggesting force coupling between the two aciculae (*Figure 6E, G*). We analysed each muscle group for their desmosomal partners and spatial orientation and inferred the possible movements of the aciculae upon

their contraction. This revealed several possible acicular movements which we termed extension, flexion, pivoting, abduction, chaetal retraction, and jostling (*Figure 6—figure supplement 4*, *Video 2*, *Video 4*). In addition, some muscles connect to epidermal cells in the parapodium and through moving these epidermal cells can indirectly also move the aciculae (i.e. 'pulled by the skin'; e.g. MUSobP-neuV) (*Video 4*).

The movement of the parapodia during crawling also requires connections to the axochord muscle (MUSax) at the midline. Animals with an ablated axochord show impared crawling indicating a structural role for this muscle (*Lauri et al., 2014*). A high density of hemidesmosomes between the axochord and the extracellular matrix dorsal to the neuronal midline agrees with this (*Figure 1—figure supplement 1B*, *Figure 1—figure supplement 2B, D*). The extracellular matrix on the midline also serves as an attachment site for the proximal ends of anterior and posterior oblique muscles and the radial-glia-like midline cells that are rich in tonofibrils.

## Acicular movements and the unit muscle contractions that drive them

The desmosomal connectome suggests that each acicula can have complex and extensive movements and the two aciculae can move independently of each other, with some coupling (e.g. through the inter-acicular muscles). To observe acicular movements in live animals, we first imaged 4-day-old crawling *Platynereis* larvae with differential interference contrast (DIC) optics. The organisation of the musculature in 3- and 4-day-old three-segmented larvae is very similar (*Figure 7—figure supplement 1*), therefore we could relate movements in the more active 4-day-old larvae to the EM data.

We used the toolbox DeepLabCut (*Mathis et al., 2018*) to train a deep residual neural network using sample video frames to learn, track, and label 26 body parts of the larvae. This allowed us to track the 12 individual aciculae and their relative angles, both to the body midline (*Figure 7A*) and to one another (*Figure 7B*). We found that the notopodial and neuropodial aciculae within one parapodium exhibit differences in movement velocity and angles relative to one another resulting in a range of angles and positions over time (*Figure 7*, *Video 5*). During the crawl cycles, the aciculae first draw inwards and forwards, diagonally towards the head, then they tilt in a posterior direction, the proximal tip of the neuropodial acicula travelling slightly faster to open up the pair, creating the larger (40–50°) inter-acicular angle (*Figure 7E–G*). This is followed by a 'piston' movement whereby the now parallel aciculae are pushed outwards, then the proximal tips tilt rostrally again, to move the chaetae back against the trunk, propelling the animal forward on that side and completing that acicular cycle.

This analysis demonstrates that the acicular pairs can have different relative positions with inter-acicular angles changing between −25° and 50° in one parapodial cycle (*Figure 7B, G*). These relative changes occur in each segment during crawling, propagating from posterior to anterior through an undulatory gait cycle (*Figure 7A*).

In order to directly visualise muscle contractions, we imaged calcium transients in larvae ubiquitously expressing GCaMP6s. Larvae held between a slide and coverslip display spontaneous contractions of different muscle groups ('twitches') revealed by increased GCaMP fluorescence. Simultaneous imaging in the DIC channel allowed us to visualise the movement of the aciculae and parapodia (*Figure 7H–J*; *Video 6*).

The contraction of the posterior ventral neuropodial acicular muscle (MUSac-neuPV) pulls the proximal end of the acicula caudally, inducing an abduction (*Figure 7H*; *Figure 6—figure supplement 4E*). These muscles connect to the proximal end of the acicula via acicular follicle cells and anchor to the basal lamina, epidermal cells and ciliated cells (paratroch) at their other end (*Figure 7K*).

The contraction of the posterior ventral neuropodial muscles (MUSobP-neuV) induces an inward movement of the distal tip of the acicula ('flexion', *Figure 6—figure supplement 4B*) leading to the alignment of the parapodia and chaetae with the longitudinal body axis (*Figure 7I*). These muscles connect to the distal end of the aciculae via acicular follicle cells and also connect to epidermal cells in the neuropodium. At their other end, MUSobP-neuV cells are anchored to the basal lamina (*Figure 7L*).

The MUSac-neuPV cells can contract simultaneously with the parapodial retractor muscles (MUSobA-re), inducing parapodial retraction and a tilt of the aciculae with their proximal tip moving rostrally (*Figure 7J*). The retractor muscles form desmosomes on chaeFC-EC (epidermal chaetal follicle) and epidermal cells at their distal end in the neuropodium and anchor to ventral midline cells and the basal

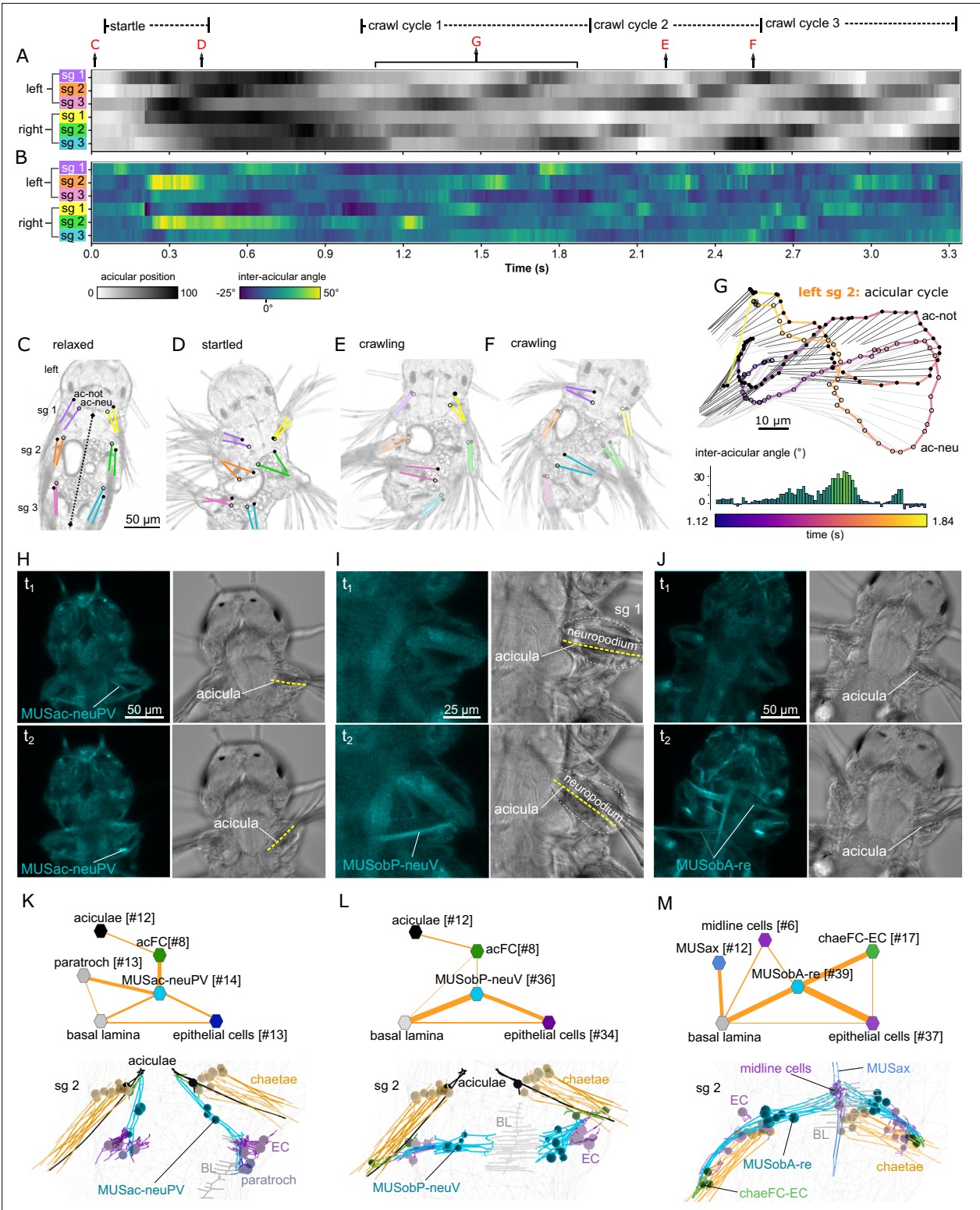

**Figure 7.** Acicula tracking, muscle contractions, and desmosomal connectivity. (**A**) Acicular position over time in all six parapodia analysed from video tracking data to indicate gait of the larva over time. Angles between the anterior–posterior bodyline connecting the mouth and the hindgut (dotted line, panel C) and each notopodial acicula were normalised (scale 0–100) within each parapodium's angular range of motion. White areas of the heatmap represent aciculae held fully back, close to the trunk. Black areas represent aciculae extended to their extreme forward position. Arrows indicate time points at which panels C–G were taken. (**B**) Inter-acicular angle between notopodial and neuropodial aciculae over time. Frames of the video sequence are labelled to show (**C**) the larva relaxed, (**D**) showing a partial startle response, (**E, F**) during crawl cycle 2 where opposite acicula pairs diverge, extend

*Figure 7 continued on next page*

*Figure 7 continued*

rostrally and inward towards the trunk, causing the larva to bend left or right. (**G**) From crawl cycle 1, acicular positions from one example acicular cycle (lasting 720 ms) are plotted for the left parapodium in segment 2, as the larva crawls forward. Points mark the proximal tips of the two aciculae, connected by coloured lines to indicate time. Finer light/dark grey straight lines show the relative positions and angles of the distally projecting aciculae. Below, the inter-acicular angles are plotted for the same parapodium over this time period. (**H**) Live imaging of MUSac-neuPV, (**I**) MUSobP-neuV, and (**J**) MUSobA-re contraction and neuropodia displacement. In H, I, and J, the left panel shows the GCaMP6s signal, the right panel shows the differential interference contrast (DIC) channel, the top and bottom panels show two frames from a video ($t_1$, $t_2$). (**K**) Desmosomal connectivity of MUSac-neuPV. Skeletons of MUSac-neuPV and their desmosomal partners in segment-2. (**L**) Desmosomal connectivity of MUSobP-neuV. Skeletons of MUSobP-neuV and their desmosomal partners in segment-2. (**M**) Desmosomal connectivity of MUSobA-re. Skeletons of MUSobA-re and their desmosomal partners in segment-2. Abbreviations: ac-not, notopodial acicula; ac-neu, neuropodial acicula; BL, basal lamina; EC, epidermal cell; sg, segment.

The online version of this article includes the following source data and figure supplement(s) for figure 7:

**Source data 1.** Source data for panel A-G of *Figure 7*.

**Figure supplement 1.** Somatic muscles in *Platynereis* larvae visualised by phalloidin Images are maximum intensity projections of CLSM-image stacks taken of phalloidin-labelled specimens.

lamina at their other end (*Figure 7M*). The ectodermal ventral midline cells bear similarities to radial glia (*Helm et al., 2017*) but lack cilia.

We could only observe these spontaneous individual muscle contractions in non-crawling larvae. During a startle response or crawling cycles, many muscles contract rapidly (*Bezares-Calderón et al., 2018*) and we were not able to spatially and temporally resolve these to individual muscle groups.

## Combined analysis of synaptic and desmosomal networks

The availability of full desmosomal and synaptic connectomes (*Verasztó et al., 2020*) for the same *Platynereis* larva allowed us to analyse how individual motoneurons could influence muscles and associated tissues. Motoneuron activation is expected to induce postsynaptic muscle contraction, which will exert forces on the desmosomal partners of the muscle. By combining desmosomal and synaptic connectomes we can infer the impact of motoneuron activation on tissue movements (*Figure 8*).

We analysed 11 motoneuron types that are well developed in the 3-day-old larva (*Bezares-Calderón et al., 2018*; *Verasztó et al., 2020*). These collectively provide broad innervation across the entire muscle network (*Figure 8A*). For this analysis, we omitted motoneurons which could not be assigned to a well-annotated cell-type category. To analyse the innervation and desmosomal connectivity of individual motoneuron types, we first plotted the synaptic pathways from the 11 motoneuron groups to the 25 out of 53 muscle groups that receive synaptic innervation (*Figure 8B*). All motoneurons have multiple muscle targets and most muscles receive synapses from more than one motoneuron.

To characterise the tissue range of each motoneuron type, we first queried their postsynaptic

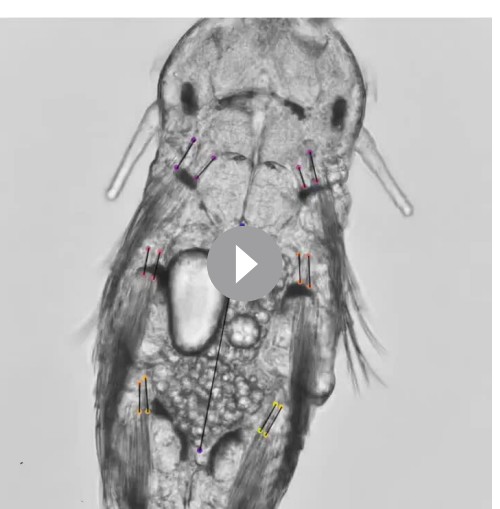

**Video 5.** Acicular movements in crawling and startling *Platynereis* larvae. Differential interference contrast (DIC) video of a crawling larva and a startling larva (end of the video) with the position of the aciculae tracked by DeepLabCut.
https://elifesciences.org/articles/71231/figures#video5

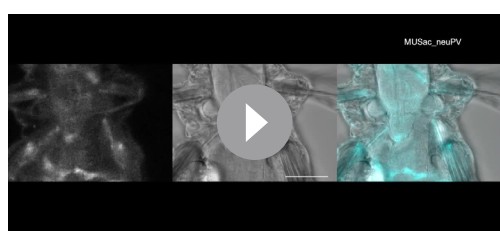

**Video 6.** Calcium imaging of muscle contractions in *Platynereis* larvae. Spontaneous contractions of individual muscle groups as visualised by the confocal imaging of calcium signals reported by GCaMP6s fluorescence. The larvae were also imaged in the differential interference contrast (DIC) channel to reveal the movements of the parapodia.
https://elifesciences.org/articles/71231/figures#video6

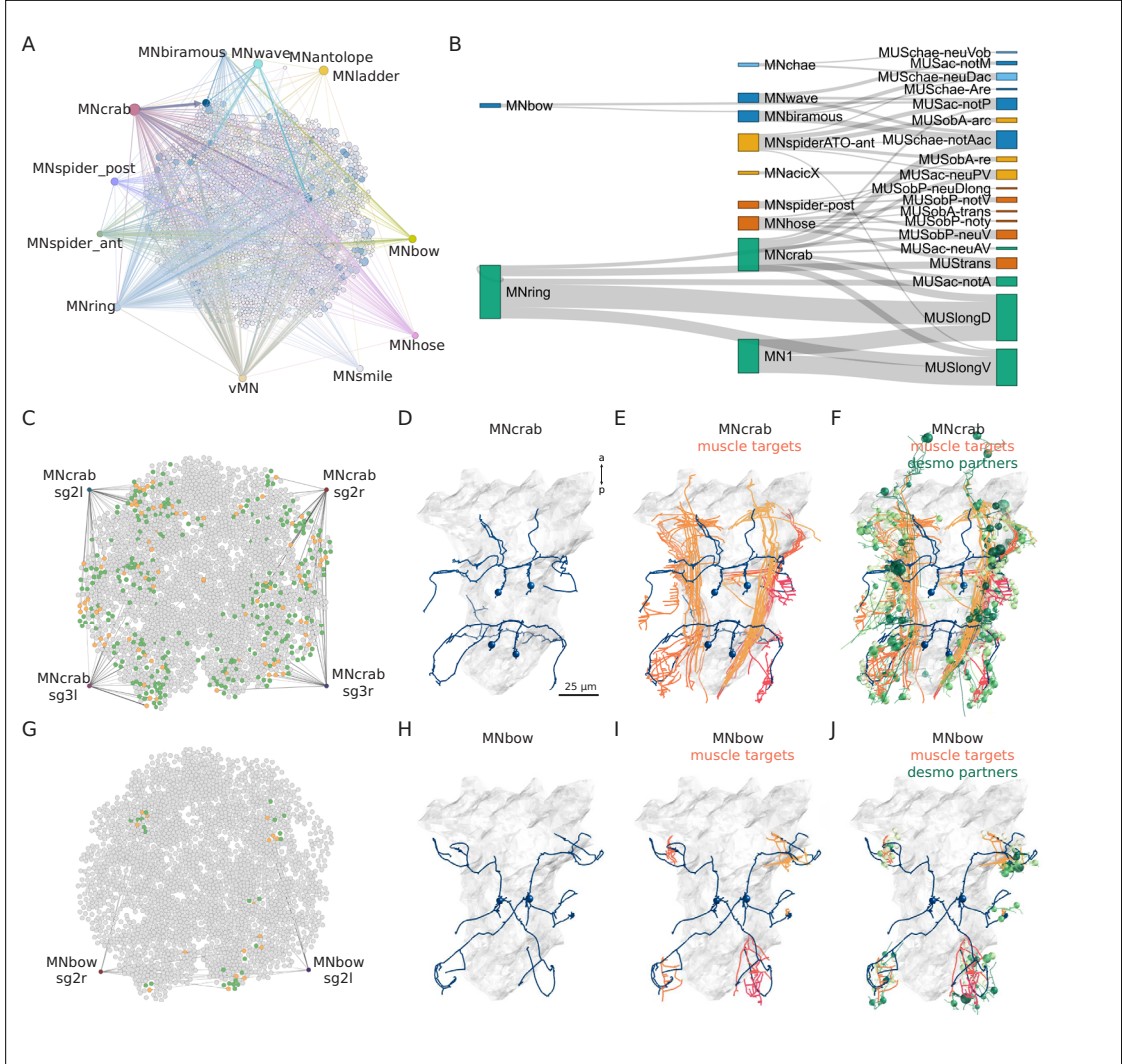

**Figure 8.** Combining synaptic and desmosomal connectomics. (**A**) Graph representation of motoneuron synaptic inputs to the desmosomal connectome. The 11 main motoneuron groups are shown. (**B**) Sankey diagram of the synaptic innervation of muscle types by motoneurons. The width of the bands is proportional to the number of synapses from a motoneuron class to a muscle class. Only connections with >9 synapses are shown. Cell types are coloured by Leiden modules determined for this grouped synaptic graph. (**C**) MNcrab motoneuron synaptic inputs to muscle cells (orange) in the desmosomal connectome and the desmosomal partners of these muscle cells (green). (**D**) Morphological rendering of the four MNcrab motoneurons. (**E**) The postsynaptic muscle partners (shades of red) of the MNcrab neurons (blue). (**F**) The postsynaptic muscle partners (shades of red) of the MNcrab neurons (blue) and the desmosomal partners of the innervated muscles (shades of green). (**G**) MNbow motoneuron synaptic inputs to muscle cells (orange) in the desmosomal connectome and the desmosomal partners of these muscle cells (green). (**H**) The two MNbow motoneurons. (**I**) The postsynaptic muscle partners (shades of red) of the MNcrab neurons (blue). (**J**) The postsynaptic muscle partners (shades of red) of the MNcrab neurons (blue) and the desmosomal partners of the innervated muscles (shades of green). In D–F and H–J, the yolk outline is shown in grey for reference. In A, C, and G the desmosomal links are not shown for clarity.

The online version of this article includes the following figure supplement(s) for figure 8:

**Figure supplement 1.** Downstream synaptic and desmosomal partners of main motoneuron types.

muscle partners in the synaptic connectome and then retrieved all desmosomal partners of these muscles (*Figure 8F and J*). We derived such combined synaptic-desmosomal graphs for 10 moto-neuron classes (*Figure 9*). Plotting the skeleton reconstructions of these cells highlights the extent of the tissue under the influence of a motoneuron class (*Figure 8C–J* and *Figure 8—figure supple-ment 1*, *Video 7*). For example, when the most highly connected muscle in the parapodial complex (MUSchae-notAac) contracts, it is expected to transmit the forces of the contraction through desmo-somes to up to nine distinct non-muscle cell types (*Figure 9*).

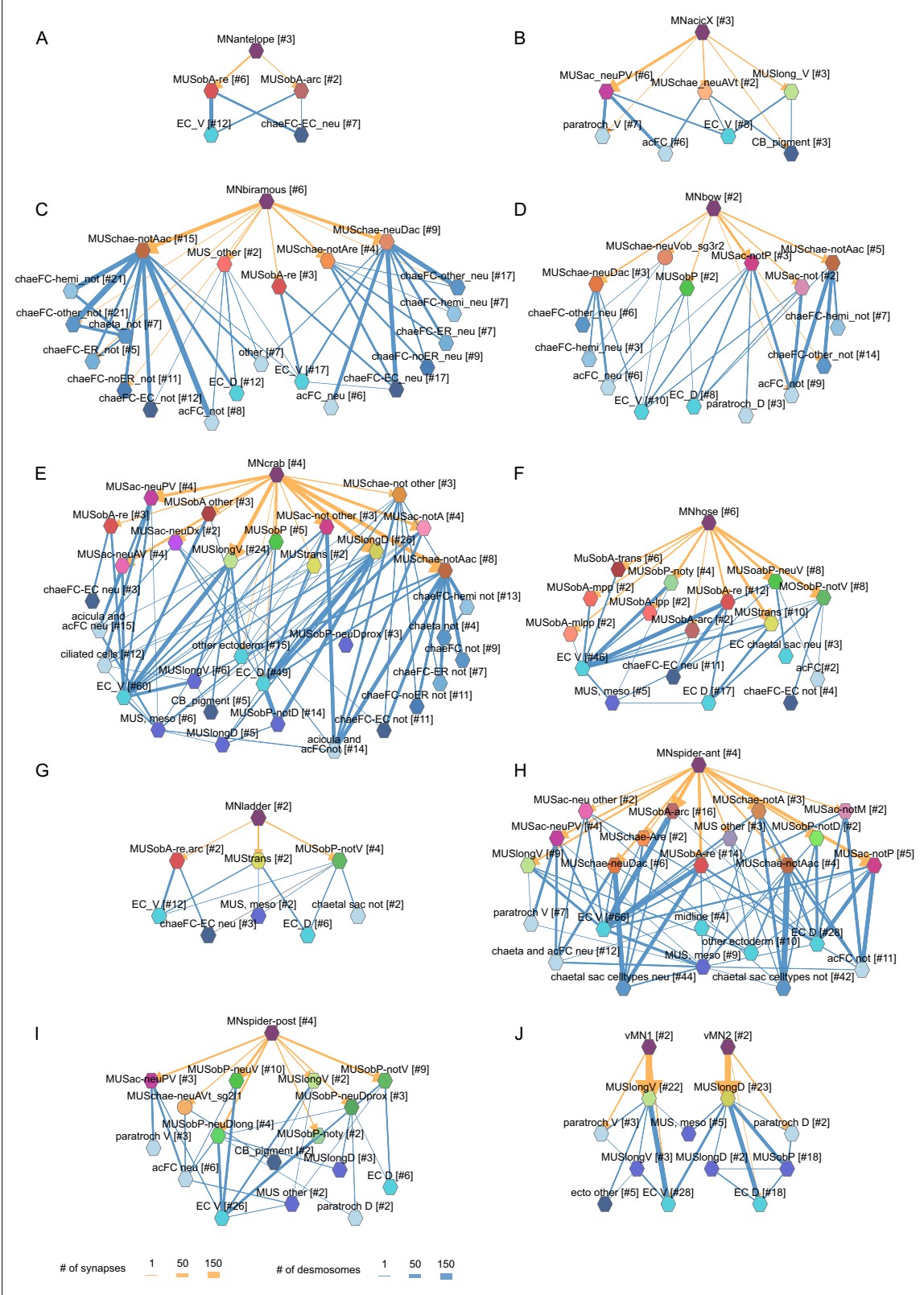

**Figure 9.** Combined synaptic–desmosomal graphs of motoneurons. Synaptic connections (orange arrows) of motoneuron classes to muscles and desmosomal links (blue edges) of the innervated muscle cells. Combined synaptic-desmosomal graph for (**A**) MNantelope, (**B**) MNacicX, (**C**) MNbiramous, (**D**) MNbow, (**E**) MNcrab, (**F**) MNhose, (**G**) MNladder, (**H**) MNspider-ant, (**I**) MNspider-post and (**J**) vMN1 and vMN2 motoneurons. Only the muscle cells directly innervated by the motoneurons are shown.

*Figure 9 continued on next page*

*Figure 9 continued*

The online version of this article includes the following source data for figure 9:

**Source data 1.** A zip archive of CATMAID json files.

Next we focused on the acicular muscle complex and highlighted each muscle and their desmosomal partners that are under the influence of a motoneuron cell type. There are eight motoneuron cell types that innervate the parapodial complex (MNcrab, MNbiramous, MNwave, MNspider-ant, MNspider-post, MNhose, MNchae, and MNring). Each of these innervates a unique combination of muscle targets – not a single muscle type (*Figure 10*). This suggests the concerted activity of anatomically distinct muscle types during the parapodial crawl cycle and other appendage movements.

## Discussion

Here, we reconstructed the somatic musculature and attached tissues in the nereid larva and showed how adhesion networks can be analysed for an entire body. We call this approach desmosomal connectomics, where the desmosomal connectome comprises all cells and extracellular structures (basal lamina, aciculae, and chaeta) and their desmosomal connections.

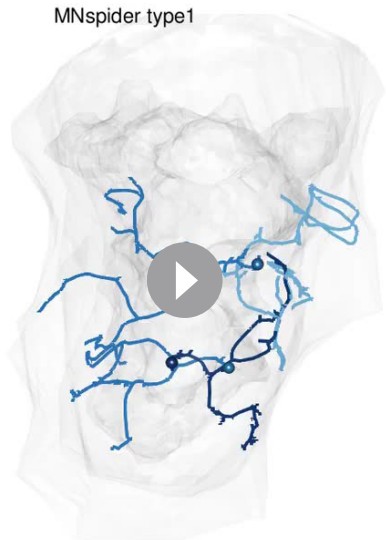

MNspider type1

**Video 7.** 3D visualisation of motoneurons, their muscle targets, and the desmosomal partners of the muscles. The first half of the video shows the combined synaptic and desmosomal connectome of different types of motoneurons. The muscles postsynaptic to the motoneurons are shown in orange. The desmosomal partners of these muscle cells are shown in green. The second half of the video shows all motoneurons from the 12 analysed motoneuron classes, their muscle partners, and the desmosomal partners of those muscles. A similar visualisation is also available in CATMAID where it can be explored interactively: https://catmaid.jekelylab.ex.ac.uk/11/links/8lzcofe. This video can be reproduced by loading the Jasek_et_al.Rproj R project in RStudio and running the code/Video7.R script (*Jasek, 2022*).

https://elifesciences.org/articles/71231/figures#video7

Our reconstructions revealed the high diversity of muscle cell types in the larva. The complexity of the nereid musculature contrasts to the relatively simple muscle architecture in the nematode *C. elegans* and the tadpole larva of the tunicate *Ciona intestinalis*. The *C. elegans* hermaphrodite has 95 body wall muscles all with a similar rhomboid shape (*Gieseler et al., 2017*). In the *C. intestinalis* larva, there are 36 muscle cells of 10 types (*Nakamura et al., 2012*). In contrast, the 3-day-old nereid larva has 853 muscle cells belonging to 53 types.

A large number of muscles engage in the movement of the parapodial muscle complex. This suggests that the diversity of nereid somatic muscles is due to the presence of an endoskeleton. Polychaete annelids are the only animals outside the tetrapods that have trunk appendages rigidified by an endoskeleton. Aciculae probably evolved in the stem group of errant annelids around the Early Ordovician (*Parry et al., 2019*; *Vinther et al., 2008*) indicating the deep ancestry of these structures, predating tetrapod limbs. In contrast to tetrapod limb bones, the aciculae are independent skeletal elements without joints. Their function is nevertheless similar in that they provide attachment sites for limb muscles and rigidity in the appendages during movement. In each parapodium, the two aciculae are moved by unique and shared sets of muscles and change their relative angle within a crawling cycle.

Beyond the complexity of the somatic musculature, our global analysis also revealed the diversity of desmosomal partners of muscle cells. These comprise over 10 different cell types, ranging from epidermal and various follicle cells through pigment cells to ciliary band cells. The stereotypy of the desmosomal connections across segments

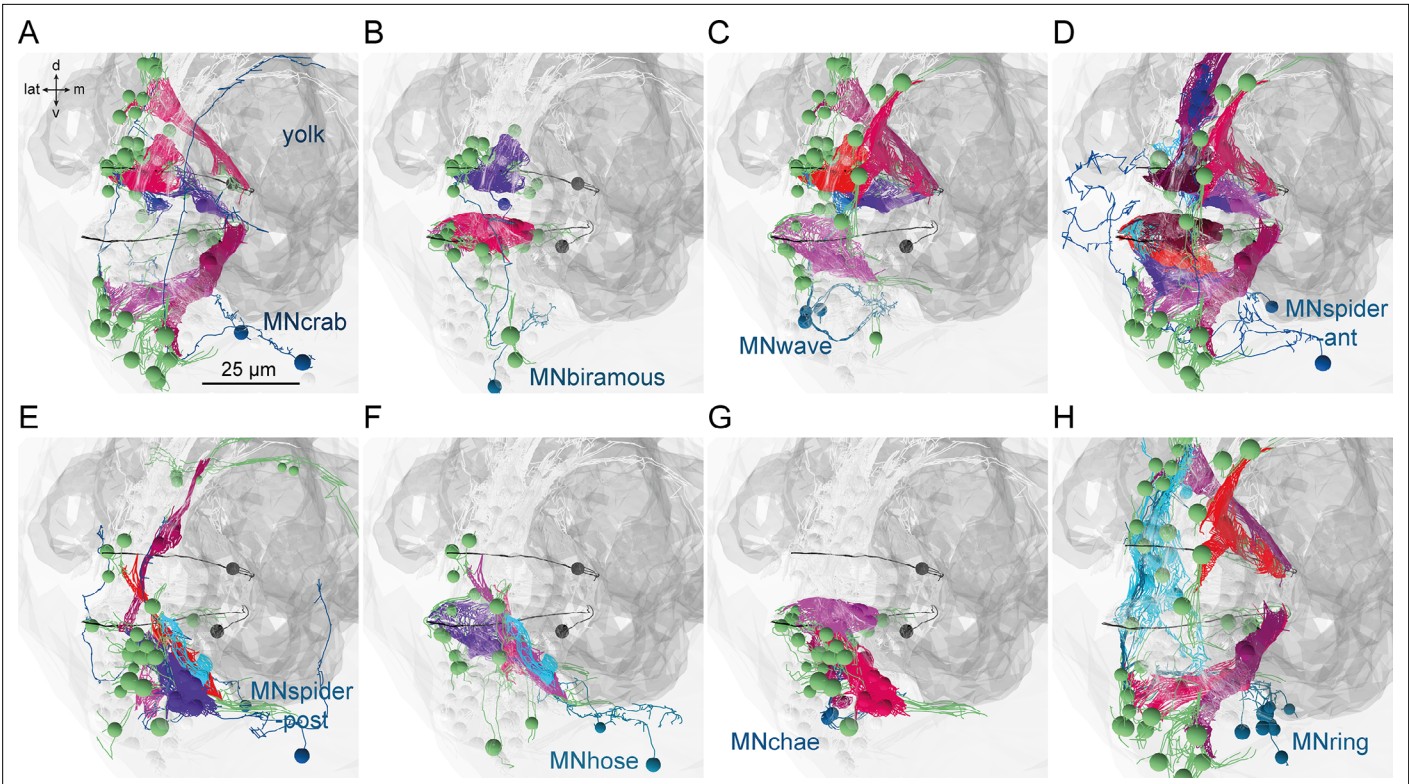

**Figure 10.** Postsynaptic muscle targets in the parapodial complex of different motoneurons. Morphological rendering of postsynaptic muscle targets (red-purple) within the parapodial complex of (**A**) MNcrab, (**B**) MNbiramous, (**C**) MNwave, (**D**) MNspider-ant, (**E**) MNspider-post, (**F**) MNhose, (**G**) MNchae, and (**H**) MNring motoneurons. The desmosomal partners of the innervated muscles are also shown (green). Only the muscles in the left parapodium of segment 2 and the motoneurons innervating this parapodium are shown.

and body sides suggests that – analogous to the synaptic connectome – the desmosomal connectome develops through a precisely specified connectivity code between cell types. Some of these connections have a clear functional relevance, as the attachment of muscles to movable chitin structures. Others are more intriguing, such as the connections to locomotor ciliary cells. These may be due to proximity and the need for muscles to attach to something, alternatively, may mediate hitherto unexplored mechanistic interactions between muscular and ciliary locomotion or represent signalling connections (*Green and Gaudry, 2000*).

The whole-body desmosomal connectome described here in combination with the synaptic connectome from the same volume (*Verasztó et al., 2020*) provide the foundation for understanding locomotor control in the nereid larva. In a crawling larva, the contraction of a large number of muscles is recruited in a precise temporal order in each limb to execute a full appendage cycle. The limb movements are organised into a gait sequence, starting from the posterior-most segment (*Figure 7*). Our imaging of spontaneous muscle twitches (*Figure 8*), prediction of unit acicular movements (*Figure 6—figure supplement 4*) and analysis of motoneuron innervation (*Figure 9*) are the first steps towards understanding parapodial movements and their neuronal control. The availability of a laboratory culture, whole-body connectome and a growing repertoire of genetic tools make the nereid larva a promising experimental system for the in-depth analysis of locomotor control.

We argue that in *Platynereis*, as well as in other animals, understanding and accurately modelling locomotion will not only require analysing nervous activity and synaptic contacts, but also comprehensive maps of the adhesive networks of muscle systems.

## Methods

### Cell reconstruction and annotations

We reconstructed all muscle and other cells by skeletonisation in CATMAID (*Saalfeld et al., 2009*). For each cell, we marked the position of the soma. Muscle cells were identified by the presence of striated myosin filaments. Developing muscles with a morphology similar to differentiated muscle cells but lacking myosin fibres were not annotated with a muscle cell type. Short segments of the basal lamina were also reconstructed as skeletons. Each basal lamina segment has at least two desmosomes and a cable length <63,700 nm. Using such relatively short and often branched fragments allowed us to only focus on basal-lamina-mediated connections for nearby cells. The basal lamina spans the entire body and runs between the ectoderm and the mesoderm. When we consider the basal lamina as one giant skeleton, it is the highest ranking node in the network and distorts the spatial layout (not shown).

We classified muscle and other cells into cell types based on their position, morphology, ultrastructural features, desmosomal connectivity, and synaptic inputs. The complete classification of neuronal cells for the same volume is described in *Verasztó et al., 2020*.

We annotated all cells by their cell-type categories (*Table 1*), the position of the soma in a body segment (episphere, segment_0, segment_1, segment_2, segment_3, pygidium), and body side (left_ side, right_side), and specific ultrastructural features (e.g. vacuolar ER, villi, dense cored vesicles, etc.). Tonofibrils were tagged as 'black fibers' in every 50th layer, or when they were encountered during tracing. Every cell containing at least one 'black fibers' tag was also annotated with the 'black fibers' annotation.

### Desmosomal connectome reconstruction

In order to mark muscle-attachment sites, we added a new connector type to CATMAID: the desmosome connector (besides synapse, gap junction, and other types). Within CATMAID, skeletonslink to each other through so-called connector nodes. Each one acts as a hub so that two skeletons can connect to each other at the same location. The type of such a link is determined by the relation attached to it. For desmosome connectors, two skeletons are allowed to connect to a connector node using the *desmosome_with* relation. We also updated various APIs and CATMAID front-end tools to support the new desmosome connector type. The connectors can then be displayed and analysed with other CATMAID widgets, including the Connectivity, the 3D View, and the Graph widgets. In Natverse, synaptic and desmosomal connectomes can also be separately analysed.

Desmosomes can be identified as dark electron-dense plaques on cell membranes with a lighter electron-dense extracellular core area (desmoglea). Intracellular intermediate filaments frequently connect to the electron-dense desmosome plaques. Hemidesmosomes were marked as connections between a cell and the basal lamina skeleton. To sample desmosomes across the body, we first sampled every 50th layer in the 4847-layer EM volume and marked every desmosome and hemidesmosome in those layers, on all cells and basal lamina. We then identified all muscle cells with less than two desmosomes from this first sampling and individually identified and marked their desmosomes.

### Network analyses

The desmosomal connectome graph was generated with the R code desmo_connectome_graph.R available in the code repository. First, we extracted all annotated desmosomes (12,666 desmosomes) from CATMAID using CATMAID's 'connectors' API endpoint. Next we retrieved all skeletons connected to the desmosome and generated a graph (2903 nodes). The largest connected component was extracted and used as the final desmosomal connectome graph (2807 nodes). Ninety-six individual nodes or nodes forming smaller clusters were removed. The final network (CATMAID annotation: desmosome_connectome) contains 2807 skeletons of which 2095 are cells with a tagged soma. The grouped graph was generated from the desmosomal graph graph by merging nodes of the same cell type into one node.

For force-field-based clustering, we exported the graph in gexf format with rgexf::igraph.to.gexf and imported it into Gephi 0.9. Force-field clustering was carried out with the Force Atlas tool in Gephi (0.9.2) The inertia was set to 0.1, repulsion strength was 35, attraction strength was 10, maximum displacement was 5, gravity was 50, speed was 5, and the 'attraction distribution' option was selected. The 'auto stabilise function' was off. Towards the end of the clustering the 'adjust by sizes' option was also selected. To prevent node overlap, we then run the 'Noverlap' function.

The new layout was exported from Gephi with normalised node coordinates as a gexf file and reimported into Rstudio with rgexf::read.gexf. The graph was then visualised with the visNetwork package with the coordinates obtained from the gexf file. Colouring was based on annotations obtained from CATMAID for each skeleton catmaid::catmaid_get_annotations_for_skeletons. The same Gephi layout could also be imported into the CATMAID Graph widget in.graphml format.

To detect modules, we used the Leiden algorithm (*Clauset et al., 2004*; *Traag et al., 2019*) (leiden::leiden method in R) with the partition type 'RBConfigurationVertexPartition' and a resolution parameter of 0.3.

To compare the desmosomal connectome to stochastic graphs, we used an R script (code/Figure4.R in the repository *Jasek, 2022*) to generate 1000 each of Erdős-Rényi, scale-free and preferential-attachment graphs with the same number of vertices and edges as the desmosomal graph. To obtain weighted graphs, we assigned the edge weights from the desmosomal graph to these stochastically generated graphs. We also generated 1000 subsamples of the desmosome graph, each with 100 nodes randomly deleted. In addition, we also generated 1000 subsamples of a reduced neuronal connectome graph from the same *Platynereis* larva (*Verasztó et al., 2020*).

## Motoneuron innervation

Motoneurons and their synaptic connections were reconstructed as described in *Verasztó et al., 2020*. No gap junctions were observed in our dataset.

## Data and code availability

All reconstructions, annotations, and EM images can be queried in a CATMAID project at https://catmaid.jekelylab.ex.ac.uk (project id: 11), together with the synaptic connectome (*Verasztó et al., 2020*). The data can also be queried by the R package Natverse (*Bates et al., 2020*). All analysis scripts used are available at https://github.com/JekelyLab/Jasek_et_al (commit b661583, *Jasek, 2022*; copy archived at swh:1:rev:b661583e56e482f7103629e5cdf23eba12813264).

## Anatomical visualisation and preparation of figures

To visualise the morphology of skeletons and soma positions in 3D, we used either the CATMAID 3D view widget or the Natverse package (*Bates et al., 2020*) in R with the rgl plot engine.

*Videos 1–4* and *Video 7* were generated in Rstudio (2022.07.01). Figures were done either in Rstudio (*Figures 1–6*, *Figure 1—figure supplements 2 and 3*, *Figure 8—figure supplement 1*) or in Adobe Illustrator (various versions) or Inkscape with some panels generated in CATMAID or Rstudio. All R code used is available at https://github.com/JekelyLab/Jasek_et_al, as a single R project (commit b661583).

## Live imaging

To image muscle contractions, we used larvae expressing GCaMP6s. Zygotes were microinjected with GCaMP6s mRNA as described (*Bezares-Calderón et al., 2018*). For imaging, we used a Zeiss LSM 880 confocal microscope with a ×40 C-Apochromat ×40/1.2 W Korr FCS M27 water-immersion objective and a 488 nm laser.

Crawling 4-day-old larvae were imaged with a Leica DMi8 microscope with DIC optics and a V1212 phantom vision research camera.

## Phalloidin staining

Three-day-old (72–77 hr after spawning) and 4-day-old (96–100 hr after spawning) larvae were relaxed in isotonic $MgCl_2$ solution (0.34 M, in distilled water) mixed 1:1 with the culture seawater. 4% PFA (paraformaldehyde in 0.1 M phosphate-buffered saline [PBS], pH 7.2, with 0.1% Tween-20) was added 1:1 to fix larvae for 15 min at room temperature (RT) on a rocking board (effective fixation concentration = 2% PFA). Specimens were washed six times for 10 min in 0.1 M PBS with 0.5% Tween-20. Muscles were labelled with 0.33 μM Alexa Fluor 633 phalloidin (Invitrogen) in 0.1 M PBS with 0.5% Triton-X and 0.025% BSA for 1.5 hr at RT on a rocking board. Incubation and all further steps were carried out in the dark. Following six 10 min washes with 0.1 M PBS, 0.5% Tween-20 specimens were mounted in Fluoromount G with DAPI (Invitrogen). At least three specimens per stage were imaged on a Leica TCS SP8 CLSM microscope with a 405-nm diode laser for DAPI and

a 633-nm HeNe laser for phalloidin labelling. We used the HyD-hybrid detectors and a ×63 or ×100 oil immersion objective. Image stacks were exported as.lif files and further analysed in Imaris (Bitplane).

## Tracking of acicular movements

We used the toolbox DeepLabCut (*Mathis et al., 2018*) to train a deep residual neural network (ResNet-50). In 30 sample video frames, we manually labelled 26 body parts of the larva, including the end of the pharynx and the developing proctodeum (posterior gut lumen), in addition to two points along the larva's 12 aciculae; the proximal tip and a more distal point. The network ran 150,000 training iterations before we evaluated it and removed outliers, making corrections for refinement. Afterwards, we merged and retrained the network to improve tracking accuracy. Two high-speed video sequences were then analysed by the trained network using this DeepLabCut toolbox. The first recording incorporated a partial startle response to physical disturbance and three complete crawling cycles. The second recording featured a complete startle response. We then created videos with labelled body parts using the filtered predictions of the network, which appear as coloured dots, some of which (e.g. proximal and distal points of the same acicula) are connected by black 'skeleton' lines. A line also connects the pharynx and proctodeum, providing a longitudinal bodyline axis (see also dotted line in *Figure 7C*). Once satisfied with the accuracy of the tracking, we analysed the *xy* positions and relative angles of aciculae from the tracking data (*Figure 7—source data 1*). To reveal patterns of gait as the larva crawled, the angles between each notopodial acicula and the larva's longitudinal axis were normalised (scale 0–100) within each acicula's own extremes of movement. This normalisation was necessary due to the different angular placements and ranges of motion between the parapodia along the segments of the body. In addition, we calculated the angles between acicular pairs within the same parapodium to assess the degree to which they remain parallel and at which times they diverge or move independently.

## Acknowledgements

We thank Liz Williams and members of the Jékely lab for comments on the manuscript. We thank Luis A Bezares-Calderón for providing GCaMP larvae. We also thank Kirsty Wan for helping with the fast DIC imaging of acicular movements. This research was supported by the FP7-PEOPLE-2012-ITN grant no. 317172 'NEPTUNE'. This research was funded by the Wellcome Trust Investigator Award 214337/Z/18/Z. This project has received funding from the European Research Council (ERC) under the European Union's Horizon 2020 research and innovation programme (grant agreement no. 101020792).

## Additional information

### Competing interests

Tom Kazimiers: Tom Kazimiers is the founder of kazmos GmbH, a company that continues the development of the open-source package CATMAID. Gáspár Jékely: Reviewing editor, *eLife*. The other authors declare that no competing interests exist.

### Funding

| Funder | Grant reference number | Author |
|---|---|---|
| European Commission | FP7-PEOPLE-2012-ITN grant no. 317172 | Sanja Jasek Gáspár Jékely |
| Wellcome Trust | Investigator Award 214337/Z/18/Z | Sanja Jasek |
| European Research Council | grant agreement No 101020792 | Alexandra Kerbl Gáspár Jékely |

| Funder | Grant reference number | Author |
|--------|------------------------|--------|

The funders had no role in study design, data collection, and interpretation, or the decision to submit the work for publication. For the purpose of Open Access, the authors have applied a CC BY public copyright license to any Author Accepted Manuscript version arising from this submission.

## Author contributions
Sanja Jasek, Conceptualization, Data curation, Software, Formal analysis, Validation, Investigation, Visualization, Methodology, Writing - original draft, Writing – review and editing; Csaba Verasztó, Data curation, Formal analysis, Validation, Investigation, Writing – review and editing; Emelie Brodrick, Data curation, Formal analysis, Investigation, Visualization, Methodology, Writing - original draft, Writing – review and editing; Réza Shahidi, Data curation, Formal analysis, Investigation, Methodology, Writing – review and editing; Tom Kazimiers, Software, Methodology, Writing – review and editing; Alexandra Kerbl, Investigation, Visualization, Methodology; Gáspár Jékely, Conceptualization, Data curation, Software, Formal analysis, Supervision, Funding acquisition, Validation, Investigation, Visualization, Methodology, Writing - original draft, Project administration, Writing – review and editing

## Author ORCIDs
Gáspár Jékely http://orcid.org/0000-0001-8496-9836

## Decision letter and Author response
Decision letter https://doi.org/10.7554/eLife.71231.sa1
Author response https://doi.org/10.7554/eLife.71231.sa2

# Additional files

## Supplementary files
• Transparent reporting form

## Data availability
All EM, tracing and annotation data are available at https://catmaid.jekelylab.ex.ac.uk. All code is available at https://github.com/JekelyLab/Jasek_et_al (copy archived at swh:1:rev:b661583e56e482f7103629e5cdf23eba12813264).

The following dataset was generated:

| Author(s) | Year | Dataset title | Dataset URL | Database and Identifier |
|-----------|------|---------------|-------------|-------------------------|
| Jasek S, Verasztó C, Shahidi R, Jékely G | 2021 | Desmosomal connectome tracing and annotation data | https://catmaid.jekelylab.ex.ac.uk | CATMAID, HT-4_Naomi |

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
