## [Editor Report]

This paper is based on the digital reconstruction of a serial EM stack of a larva of the annelid *Platynereis* and presents a complete 3D map of all desmosomes between somatic muscle cells and their attachment partners. This resource is of interest to scientists in several fields: motor control, high-resolution anatomy, and network analyses. With the first comprehensive and complete mapping of muscle-to-body connectivity through desmosomes in an annelid larva, it has the potential to close a missing link and make progress towards understanding in a "holistic" way how a complex neural circuitry controls an equally complex pattern of movement/behavior.

---

## [Decision Letter]

**Decision letter after peer review:**

Thank you for submitting your article "Desmosomal connectomics of all somatic muscles in an annelid larva" for consideration by *eLife*. Your article has been reviewed by 3 peer reviewers, and the evaluation has been overseen by a Reviewing Editor and Marianne Bronner as the Senior Editor. The reviewers have opted to remain anonymous.

All reviewers were impressed by the large dataset and agree that the paper is in principle suitable as a resource article for *eLife*. However, there had been some discussions on the required extent of data re-analysis, ultimately connected to the question what kind of biological insights might be possible to answer with this dataset as it stands and what is important for biological meaningful insights in the future. Based on this discussion, we concluded on the following requests:

1. Adding information on the presence/absence of gap junctions. This would provide important information on the possible cellular communication.

2. Adding the consideration of the vector-like nature of muscle cells and the likely different functions of desmosomes depending on their relative position in the muscle cell to the manuscript. The suggestion is to filter your data on the level of each muscle cell, which subsequently should allow to tell apart three regions: front and back end, where the fibers exert contraction force, and sides, where desmosomes in-between fibers (or between fibers and epidermis) somehow stabilize the muscle or bodywall. This should provide important detailed information as to how the muscles could exert their forces.

There had been some discussion as to how feasible the addition of the requested information of points 1/2 is within a reasonable timeframe. If this appears to be an unreasonable request, we would ask the authors to get back to us with a suggestion how they could discuss this limitation of their data analysis.

3. improved visualisation of the data, especially details on the acicular movements:

The descriptions of the aciculae are difficult to follow. A simulation/video showing how the respective muscle groups lead to different movements of the acicula in 4D or a detailed annotation of Video 4 would be highly informative. Please also add information on where (relative to the acicular base) the epithelial cell (or point in the basal lamina) is located.

Please provide all annotations to a level that is understandable to the interested non-annelid expert.

4. Reformulations of different parts of the manuscript have been requested by all reviewers. There are two rationales behind this request.

A) Improvement of the understandability. Please see the individual "recommendation for the authors":

Reviewer 2: see "recommendation for the authors"

Reviewer 3: questions mentioned under point 1

B) A clearer vision of what kind of biologically relevant questions can be answered with the dataset in the future.

*Reviewer #1 (Recommendations for the authors):*

The timeliness, depth and quality of analysis and importance towards a full-body understanding of locomotion makes this manuscript an important and interesting work for scientists of a range of different fields, and I therefore think that is highly suited for publication in *eLife*. The manuscript generally reads well, although some parts would benefit from more precise formulations, clarification and editing such as restructuring or removing technical details.

– It would be interesting to know if muscle cells are not only physically connected by desmosomes but also electrically by gap junctions. It would be interesting if the authors could add data/information about this point to the manuscript.

– I find the possible acicular movements ('abduction,…'p.8, last paragraph and Figure 6-supp3) very difficult to follow. One problem might be that in Figure 6-supp3, the body axes are mostly missing. In addition, it is inherently difficult to understand spatial and temporal changes using a two-dimensional schematic. If possible, a simulation/video showing how the respective muscle groups lead to different movements of the acicula would be highly informative.

– Related to the previous point, I have difficulties to follow the relative 4-dimensional movements of the aciculae described in Figure 7G, and a 4D animation (if possible) of the relative movements of the aciculae would certainly help the reader. Alternatively, annotating Video 4 might help understanding acicular movements better.

*Reviewer #2 (Recommendations for the authors):*

1) This paper is rich in data and anatomical details, and would be of benefit to scientists who primarily work on other model systems. An obstacle to this goal is that the current draft is dense, making it a slow read (I learned a lot, after several goes). Reorganization can improve this article's readability for a broader audience:

– Some jargon-like technical details distract readers from focusing on main ideas. I would relegate them to Methods (Example: how authors use CATMAID and add a new function in CATAMID to annotate desmosome).

– This article might assume that readers have some foreground knowledge on both basic anatomy of this animal model and on network theory – not a large community. I thus would appreciate the results to be explained in an intuitive way. For an example, this animal model's overall anatomy should be introduced foremost in the main text. Figure 1A was a nice illustration, but with no explanation in the results, and the single line of figure legend does not offer any information either (I searched online to find some fascinating details in the differences between sg0, sg2, and sg3 that may be relevant to results in later sections).

Including such details will allow readers to put those graphs in their anatomic context naturally, instead of having a reader (e.g. me) need to research each anatomic detail as the name popped up in the text.

– Similarly, all figure legends are exceptionally brief and generally do not provide sufficient guides for the non-expert to understand the results. For example, many figures use CATMAID's output. I believe that circles in most graphs represent the soma of cells, hence the large vs small circles (e.g. in Figure 1E) reflect their size difference. However, this was not explained anywhere, and a general audience would not be able to deduce such information.

2) Below are a few technical questions and discrepancies that authors should address:

– The authors refer to their connectivity map as the 'desmosome connectome'. I think both desmosome and hemidesmosome are included in the final dataset. If so, this needs to be clarified in the sentence where they first defined the term 'desmosome connectome'.

– Page 4: 'The largest cluster of interconnected nodes consists of 2,506 nodes.' I find this confusing. If the total number of nodes is 2,524, this sentence simply means that almost all cells are inter-connected by desmosomes. Is this so? If so, this is not reflected in Figure 1E.

– Page 6: 'The largest Eigenvalue of a graph is another key network property influencing dynamic processes (Restrepo et al. 2006). This value is also largest for the desmosomal graph.'

The authors should provide some context to this and all other graph theory terminology in this section; 'dynamic processes', 'radius of network' etc. appear vague and abstract wordings for a few simple operations that could be explained plainly and interpretated intuitively. E.g. For the second part of the example sentence, I think the authors meant 'The largest Eigenvalue of weighted (?) adjacency matrix of the desmosomal connectivity map'?

– Page 7: 'Nodes corresponding to the neuropodial and notopodial parts of the parapodia also occupy distinct domains (Figure 5A-F).' In Figure 5D, at least sg0 cells appear more much distributed than their physical locations (5C)? The authors should provide more details to explain what is shown in figures.

– Page 4-6: Section 'Local connectivity and modular structure of the desmosomal connectome'. The authors applied an improved clustering algorithm (Leiden) to assign nodes of the biological desmosomal connectome, but later used the Louvain algorithm to compare the modularity scores for their subsampled connectomes with that random networks. The same algorithm should be used to assess and compare community structures.

*Reviewer #3 (Recommendations for the authors):*

This paper is based on digital reconstruction of a serial EM stack of a larva of the annelid *Platynereis* and presents a complete 3D map of all desmosomes between somatic muscle cells and their attachment partners, including muscle cells, glia, ciliary band cells, epidermal cells and specialized epidermal cells that anchor cuticular chaetae (circumchaetal cells) and aciculae (circumacicular cells). The rationale is that the spatial patterning of desmosomes determines the direction of forces exerted by muscular contraction on the body wall and its appendages will determine movement of these structures, which in turn results in propulsion of the body as part of specific behaviors.

To go a step further, if connecting this desmosome connectome with the (previously published) synaptic connectome, one may gain insight into how a specific spatio-temporal pattern of motor neuron activity will lead, via a resulting pattern of forces caused by muscles, to a specific behavior. In the authors' words: "By combining desmosomal and synaptic connectomes we can infer the impact of motoneuron activation on tissue movements". This is an interesting idea which has the potential to make progress towards understanding in a "holistic" way how a complex neural circuitry controls an equally complex behavior. The analysis of the EM data appears solid; the authors can show convincingly that desmosomes can be resolved in their EM dataset; and the technology used to plot and analyze the data is clearly up to the task. My main concern is with the way in which the desmosome pattern is entered in the analysis, which I think makes it very difficult to extract enough relevant information from the analysis that would reach the stated goal.

1. The context of how different structures of the *Platynereis* larval body, by changing their position, move the body needs much more introduction than the short paragraph given at the end of the Introduction.

– My understanding is that the larval body is segmented, and contraction of the segments can cause a certain type crawling or swimming: does it? Do the longitudinal muscles, for example, insert at segment boundaries, and alternating contraction left-right cause some sort of "wiggling" or peristalsis?

– In addition, there are segmental processes (parapodia, neuropodia), and embedded in them are long chitinous hairs (Chaetae, Acicula). Do certain types of the muscles described in the study insert at the base of the parapodia/neuropodia (coming from different angles), such that contraction would move the entire process, including the chaetae/acicula embedded in their tips?

– Or is it that only these chaetae/acicula move, by means of muscles inserting at their base (the latter is clearly part of the story)? Or does both happen at the same time: parapodium moves relative to the trunk, and chaeta/acicula moves relative to the parapodium? How would these movements lead to different kind of behaviors?

– Diagrams should be provided that shed light on these issues.

2. The main problem I have with the analysis is the way a muscle cell is treated, namely as a "one dimensional" node, rather than a vector.

– In the current state of the analysis, the authors have mapped all desmosomes of a given muscle cell to its attached "target" cell. But how is that helpful? The principal way a muscle cell acts is by contracting, thereby pulling the cells it attaches to at its two end closer together. As the authors state (p.4) "…desmosomes..are enriched at the ends of muscle cells indicating that these adhesive structures transmit force upon muscle-cell contraction."

– for that reason, the desmosomes at the muscle tips have to be treated as (2) special sets. Aside from these tip desmosomes there are other desmosomes (inbetween muscles, for example), but they (I would presume) have a very different function; maybe to coordinate muscle fiber contraction? Augment the force caused by contraction?

– As far as I understand for (all of) the desmosome connectome plots, there is no differentiation made between desmosome subsets located at different positions within the muscle fiber. I therefore don't see how the plots are helpful to shed light on how the multiplicity of muscles represented in the graphs cause specific types of neurons.

– As it stands these plots "merely" help to classify muscles, based on their position and what cell type they target: but that (certainly useful) map could have probably also be achieved by light microscopic analysis.

3. Section "Local connectivity and modular structure of the desmosomal connectome" p.4-7" undertakes an analysis of the structure of the desmosome network, comparing it with other networks.

– What is the rationale here? How do the conclusions help to understand how the spatial pattern of muscles and their contraction move the body?

– Isn't, on the one hand (given that position of the desmosome was apparently not considered), the finding that desmosome networks stand out (from random networks) by their high level of connectivity ("with all cells only connecting to cells in their immediate neighbourhood forming local cliques") completely expected?

– On the other hand, does this reflect the reality, given that (many?) muscle cells are quite long, connecting for example the anterior border of a segment with the posterior border.

4. In the section "Acicular movements and the unit muscle contractions that drive them" the authors record movement of the acicula and correlate it with activity (Ca imaging) of specific muscle types. This study gives insightful data, and could be extended to all movements of the larva.

– The fact that a certain muscle is active when the acicula moves in a certain direction can be explained (in part) by the "connectivity": as shown in Figure 7L, the muscle inserts at a circumacicular cell on the one side, and to an epithelial (epidermal?) cell and the basal lamina on the other side. But how meaningful is a description at this "cell type level" of resolution? The direction of acicula deflection depends on where (relative to the acicula base) the epithelial cell (or point in the basal lamina) is located. This information is not given in the part of the connectome network shown in Figure 7L, or any of the other graphs.

---

## [Author Response]

Reviewer #1 (Recommendations for the authors):– It would be interesting to know if muscle cells are not only physically connected by desmosomes but also electrically by gap junctions. It would be interesting if the authors could add data/information about this point to the manuscript.

We have not observed gap junctions in our dataset. We updated the Methods section to mention the absence of observed gap junctions.

– I find the possible acicular movements ('abduction,…'p.8, last paragraph and Figure 6-supp3) very difficult to follow. One problem might be that in Figure 6-supp3, the body axes are mostly missing. In addition, it is inherently difficult to understand spatial and temporal changes using a two-dimensional schematic. If possible, a simulation/video showing how the respective muscle groups lead to different movements of the acicula would be highly informative.

We have added the body axes to this figure (Figure6—figure supplement 3) and simplified it to only show those movements that we could infer confidently. We also included a 3D animation, which shows different muscle groups from different angles (Video 2). This should help to see the different muscle groups relative to the aciculae in the 3D volume.

– Related to the previous point, I have difficulties to follow the relative 4-dimensional movements of the aciculae described in Figure 7G, and a 4D animation (if possible) of the relative movements of the aciculae would certainly help the reader. Alternatively, annotating Video 4 might help understanding acicular movements better.

We have annotated Video 4 to show which are the notopodial and the neuropodial aciculae in the different segments. Our live imaging data did not allow us to track the 4-dimensional movements of the aciculae because we have limited information along the Z-axis. We could tell which chaetae and aciculae are more ventral based on the focus. Tracking the 4D movement of the aciculae would require fast volumetric imaging that we were not able to do in the scope of this project.

Reviewer #2 (Recommendations for the authors):1) This paper is rich in data and anatomical details, and would be of benefit to scientists who primarily work on other model systems. An obstacle to this goal is that the current draft is dense, making it a slow read (I learned a lot, after several goes). Reorganization can improve this article's readability for a broader audience:– Some jargon-like technical details distract readers from focusing on main ideas. I would relegate them to Methods (Example: how authors use CATMAID and add a new function in CATAMID to annotate desmosome).

We thank the reviewer for these comments. We have moved some of the technical details to the methods section. We have also shortened the Introduction and simplified the text describing the statistics of the desmosomal connectome.

– This article might assume that readers have some foreground knowledge on both basic anatomy of this animal model and on network theory – not a large community. I thus would appreciate the results to be explained in an intuitive way. For an example, this animal model's overall anatomy should be introduced foremost in the main text. Figure 1A was a nice illustration, but with no explanation in the results, and the single line of figure legend does not offer any information either (I searched online to find some fascinating details in the differences between sg0, sg2, and sg3 that may be relevant to results in later sections).Including such details will allow readers to put those graphs in their anatomic context naturally, instead of having a reader (e.g. me) need to research each anatomic detail as the name popped up in the text.

We have added more detail about the anatomy of the larva to the Introduction:

“In *Platynereis*, muscle development starts in the planktonic, ciliated trochophore larval stage (one to two days old) (Fischer et al., 2010). The older (around three to six days) nectochaete larvae have three main trunk segments, each with a pair of appendages called parapodia. The parapodia are composed of a ventral lobe (neuropodium) and a dorsal lobe (notopodium) and each lobe contains a single acicula and a bundle of chitin bristles (chaetae)(Hausen, 2005). *Platynereis* larvae have an additional cryptic segment between the head and the main trunk segments (Steinmetz et al., 2011). This cryptic segment (also referred to as segment 0) lacks parapodia and gives rise to the first pair of tentacular cirri that start to grow around three days of development.”

We also changed the colouring of the muscles in Figure 1B to highlight the segments more clearly.

– Similarly, all figure legends are exceptionally brief and generally do not provide sufficient guides for the non-expert to understand the results. For example, many figures use CATMAID's output. I believe that circles in most graphs represent the soma of cells, hence the large vs small circles (e.g. in Figure 1E) reflect their size difference. However, this was not explained anywhere, and a general audience would not be able to deduce such information.

We have extended the figure legends including an explanation of what smaller vs larger spheres represent in the morphological renderings.

“Morphological rendering of traced skeletons and soma position (spheres) of all acicular muscles and their classification.”

“ Position and size of cell nuclei, represented as spheres, of all the cells…”.

2) Below are a few technical questions and discrepancies that authors should address:– The authors refer to their connectivity map as the 'desmosome connectome'. I think both desmosome and hemidesmosome are included in the final dataset. If so, this needs to be clarified in the sentence where they first defined the term 'desmosome connectome'.

We have clarified the definition of the desmosomal connectome.

“These desmosomal connections form a body-wide network through which tensile forces propagate. This network comprises all muscles and all their cellular and non-cellular (e.g., basal lamina) partners. We refer to this network as the desmosomal connectome, with muscles, other cells and basal lamina chunks as nodes and hemidesmosomes and desmosomes as links.”

– Page 4: 'The largest cluster of interconnected nodes consists of 2,506 nodes.' I find this confusing. If the total number of nodes is 2,524, this sentence simply means that almost all cells are inter-connected by desmosomes. Is this so? If so, this is not reflected in Figure 1E.

Almost all cells which have desmosomes are interconnected into a single graph either directly or indirectly through the basal lamina. We clarified this in the text. We also updated Figure 1E (now 1F) to show the edges more clearly.

– Page 6: 'The largest Eigenvalue of a graph is another key network property influencing dynamic processes (Restrepo et al. 2006). This value is also largest for the desmosomal graph.'The authors should provide some context to this and all other graph theory terminology in this section; 'dynamic processes', 'radius of network' etc. appear vague and abstract wordings for a few simple operations that could be explained plainly and interpretated intuitively. E.g. For the second part of the example sentence, I think the authors meant 'The largest Eigenvalue of weighted (?) adjacency matrix of the desmosomal connectivity map'?

We have thoroughly revised this section, see also our responses to Reviewer #1. We simplified the terminology and discuss fewer network indicators that best highlight the differences between the different types of graphs.

– Page 7: 'Nodes corresponding to the neuropodial and notopodial parts of the parapodia also occupy distinct domains (Figure 5A-F).' In Figure 5D, at least sg0 cells appear more much distributed than their physical locations (5C)? The authors should provide more details to explain what is shown in figures.

We now describe in more detail some cells that are more distributed in the graph than their physical locations.

“Exceptions include the dorsolateral longitudinal muscles (MUSlongD) in segment-0. These cells connect to dorsal epidermal cells that also form desmosomes with segment-1 and segment-2 MUSlongD cells. These connections pull the MUSlongD_sg0 cells down to segment-2 in the force-field layout (Figure 5D).”

– Page 4-6: Section 'Local connectivity and modular structure of the desmosomal connectome'. The authors applied an improved clustering algorithm (Leiden) to assign nodes of the biological desmosomal connectome, but later used the Louvain algorithm to compare the modularity scores for their subsampled connectomes with that random networks. The same algorithm should be used to assess and compare community structures.

We originally used the Louvain algorithm because the Leiden method is computationally ~100 times more intensive. We have now rerun the computations with the Leiden method and updated the relevant code and data. The desmosomal connectome has the highest modularity value also when calculated with the Leiden algorithm, therefore our original conclusions did not change.

Reviewer #3 (Recommendations for the authors):This paper is based on digital reconstruction of a serial EM stack of a larva of the annelid Platynereis and presents a complete 3D map of all desmosomes between somatic muscle cells and their attachment partners, including muscle cells, glia, ciliary band cells, epidermal cells and specialized epidermal cells that anchor cuticular chaetae (circumchaetal cells) and aciculae (circumacicular cells). The rationale is that the spatial patterning of desmosomes determines the direction of forces exerted by muscular contraction on the body wall and its appendages will determine movement of these structures, which in turn results in propulsion of the body as part of specific behaviors.To go a step further, if connecting this desmosome connectome with the (previously published) synaptic connectome, one may gain insight into how a specific spatio-temporal pattern of motor neuron activity will lead, via a resulting pattern of forces caused by muscles, to a specific behavior. In the authors' words: "By combining desmosomal and synaptic connectomes we can infer the impact of motoneuron activation on tissue movements". This is an interesting idea which has the potential to make progress towards understanding in a "holistic" way how a complex neural circuitry controls an equally complex behavior. The analysis of the EM data appears solid; the authors can show convincingly that desmosomes can be resolved in their EM dataset; and the technology used to plot and analyze the data is clearly up to the task. My main concern is with the way in which the desmosome pattern is entered in the analysis, which I think makes it very difficult to extract enough relevant information from the analysis that would reach the stated goal.1. The context of how different structures of the Platynereis larval body, by changing their position, move the body needs much more introduction than the short paragraph given at the end of the Introduction.– My understanding is that the larval body is segmented, and contraction of the segments can cause a certain type crawling or swimming: does it? Do the longitudinal muscles, for example, insert at segment boundaries, and alternating contraction left-right cause some sort of "wiggling" or peristalsis?

Longitudinal muscles do not insert only at segment boundaries, but have desmosomal connections along the entire length of the cell. Individual longitudinal muscle cells can span up to 3 segments. However the cells are staggered in such a way that all longitudinal muscle cells with somas in one segment can collectively cover up to 4 segments. Longitudinal muscles are involved in turning when swimming (Randel et al., 2014). The undulatory trunk movements and parapodial walking movements are due to the contraction of oblique and parapodial muscles. The longitudinal muscles provide support during crawling (via desmosomal links) but it is unlikely that these muscles contract segmentally. Disentangling the distinct contributions of 53 types of muscles during crawling will require further studies.

– In addition, there are segmental processes (parapodia, neuropodia), and embedded in them are long chitinous hairs (Chaetae, Acicula). Do certain types of the muscles described in the study insert at the base of the parapodia/neuropodia (coming from different angles), such that contraction would move the entire process, including the chaetae/acicula embedded in their tips?

Yes, acicular muscles insert at the proximal base of the acicula, and by moving the acicula they move the entire noto-/neuropodia. We have presented the anatomy of all acicular and chaetal muscles types in the figures and videos.

– Or is it that only these chaetae/acicula move, by means of muscles inserting at their base (the latter is clearly part of the story)? Or does both happen at the same time: parapodium moves relative to the trunk, and chaeta/acicula moves relative to the parapodium? How would these movements lead to different kind of behaviors?– Diagrams should be provided that shed light on these issues.

We have extended Video 2 to show individual muscles and their relation to the aciculae in one of the parapodia. We also clarified this in the text:

“Several acicular muscles attach on one end to the proximal base of the aciculae and on the other end to the paratrochs and epidermal cells. Oblique muscles attach to the basal lamina, epidermal and midline cells at their proximal end, run along the anterior edge of parapodia and attach to epidermal and chaetal follicle cells at their distal tips. Both of these muscle groups are involved in moving the entire parapodium. Acicular muscles move the proximal tips of the aciculae, while oblique muscles move the parapodium by moving the tissue around the chaetae and the aciculae. All acicular movements also correspond to parapodial movements. Chaetae are embedded in the parapodium and therefore move with it, but the chaetal sac muscles can also independently retract the chaetae into the parapodium or protract them and make them fan out.”

2. The main problem I have with the analysis is the way a muscle cell is treated, namely as a "one dimensional" node, rather than a vector.– In the current state of the analysis, the authors have mapped all desmosomes of a given muscle cell to its attached "target" cell. But how is that helpful? The principal way a muscle cell acts is by contracting, thereby pulling the cells it attaches to at its two end closer together. As the authors state (p.4) "…desmosomes..are enriched at the ends of muscle cells indicating that these adhesive structures transmit force upon muscle-cell contraction."

At the level of the current analysis our data reveal which cells may be moved by the contractions of the individual muscle cells. The reviewer is right that treating a muscle as a vector (or set of vectors) would be a more accurate description, which would potentially also open up the possibility of computational modelling. We have provided such a vectorised dataset in the revised version, where each muscle-cell skeleton is subdivided into short linear segments (Figure2–source–data 2). This dataset may be useful to approach the problem with a three dimensional approach, which is beyond the scope of the current analysis. We also included an additional video (Video 7) showing examples of muscles and their partners where the cells and the desmosomes connecting them are highlighted. This reveals that the desmosomes connecting two cells are often at the very end of the muscle cell.

– for that reason, the desmosomes at the muscle tips have to be treated as (2) special sets. Aside from these tip desmosomes there are other desmosomes (inbetween muscles, for example), but they (I would presume) have a very different function; maybe to coordinate muscle fiber contraction? Augment the force caused by contraction?

Desmosomes between muscles only occur between muscles of different types, not for homotypic connections. There are other types of junctions (adhaerens-like junctions) that connect individual cells of a muscle bundle together (not analysed here). We clarified this in the text.

– As far as I understand for (all of) the desmosome connectome plots, there is no differentiation made between desmosome subsets located at different positions within the muscle fiber. I therefore don't see how the plots are helpful to shed light on how the multiplicity of muscles represented in the graphs cause specific types of neurons.

We would like to point out that the cells and structures that muscles connect to via desmosomes are very likely the parts of the body that will move during the contraction of the muscle or will provide structural support (e.g. basal lamina) for the muscle cell to contract. This is most evident in the parapodial complex. The majority of muscles in the body connect to the aciuclar folliclecells and the aciculae are the most actively moving parts in the body during crawling (see Video 4). In any case, since we provide all skeleton reconstructions and the xyz coordinates of all desmosomes, the data could be further analysed following these suggestions by the reviewer.

– As it stands these plots "merely" help to classify muscles, based on their position and what cell type they target: but that (certainly useful) map could have probably also be achieved by light microscopic analysis.

This has never been achieved by light microscopy analysis in the hundreds of papers on invertebrate muscle anatomy (e.g. by phalloidin staining). For an LM analysis, it would not be sufficient to label the muscle fibres, but one would also need to label the desmosomes and a multitude of non-muscle cell types including the extent of their cytoplasm. This is technically very challenging (we would nevertheless be happy to hear specific suggestions for markers etc. from the Reviewer). Currently, only EM provides the required depth of structural information and resolution. This is why we believe that our dataset and analysis is unique, despite over a century of research in invertebrate anatomy.

3. Section "Local connectivity and modular structure of the desmosomal connectome" p.4-7" undertakes an analysis of the structure of the desmosome network, comparing it with other networks.– What is the rationale here? How do the conclusions help to understand how the spatial pattern of muscles and their contraction move the body?

We hope that our analysis may also be of interest to the community of network scientists and we believe that the reconstruction of a quite large and novel type of biological network warrants a more quantitative network analysis, using the standard methods and measures of network science – as we presented e.g. in Figure 4 – even if these mathematical analyses may not directly reveal how muscles move the body. We hope that some readers with an interest in quantitative analyses will also appreciate the broader picture here.

– Isn't, on the one hand (given that position of the desmosome was apparently not considered), the finding that desmosome networks stand out (from random networks) by their high level of connectivity ("with all cells only connecting to cells in their immediate neighbourhood forming local cliques") completely expected?

We disagree that the result was completely expected. Even if this was the case, we think it is quite different to say that a result is expected or to thoroughly quantify certain parameters and mathematically characterise key properties of the desmosomal graph (as we have done). These network analyses help to conceptualise our findings and to think about the muscle system in more global, whole-body terms.

– On the other hand, does this reflect the reality, given that (many?) muscle cells are quite long, connecting for example the anterior border of a segment with the posterior border.

Indeed, a quantitative analysis helped us to identify cases where the reality deviated somewhat from what was completely expected, and we thank the reviewer for these comments. As we explain in the revised version, some longitudinal muscles show an unexpected position in the force-field layout of the graph, due to their long-range connections. We have added extra clarifications to the text:

“To analyse how closely the force-field-based layout of the desmosomal connectome reflects anatomy, we coloured the nodes in the graph based on body regions (Figure 5). In the force-field layout, nodes are segregated by body side and body segment. Exceptions include the dorsolateral longitudinal muscles (MUSlongD) in segment-0. These cells connect to dorsal epidermal cells that also form desmosomes with segment-1 and segment-2 MUSlongD cells. These connections pull the MUSlongD_sg0 cells down to segment-2 in the force-field layout (Figure 5D).”

4. In the section "Acicular movements and the unit muscle contractions that drive them" the authors record movement of the acicula and correlate it with activity (Ca imaging) of specific muscle types. This study gives insightful data, and could be extended to all movements of the larva.– The fact that a certain muscle is active when the acicula moves in a certain direction can be explained (in part) by the "connectivity": as shown in Figure 7L, the muscle inserts at a circumacicular cell on the one side, and to an epithelial (epidermal?) cell and the basal lamina on the other side. But how meaningful is a description at this "cell type level" of resolution? The direction of acicula deflection depends on where (relative to the acicula base) the epithelial cell (or point in the basal lamina) is located. This information is not given in the part of the connectome network shown in Figure 7L, or any of the other graphs.

This information is indeed not shown in the graphs, where each cell is treated as a node. However, we provide this information in the detailed anatomical figures in Figure 6 —figure supplement 1-3 and Video 7, where the individual acicular and oblique muscle types are visualised. In principle, one could subdivide aciculae into e.g. proximal and distal halves and derive a more detailed network. We have not done this but since all the EM, anatomical rendering and connectivity data are available in our public CATMAID server (https://catmaid.jekelylab.ex.ac.uk/), we hope that the interested readers will be able to further analyse the data.

We renamed ‘epithelial’ cells to ‘epidermal’ cells.